# CAN TRANSFORMERS BE TREATMENT EFFECT ESTIMATORS?

## ABSTRACT

In this paper, we develop a general framework for leveraging Transformer architectures to address a variety of difficult treatment effect estimation (TEE) problems. Our methods are applicable both when covariates are tabular and when they consist of sequences (e.g., in text), and can handle discrete, continuous, structured, or dosage-associated treatments. While Transformers have already emerged as dominant methods for diverse domains, including natural language and computer vision, our experiments with **Trans**formers as **T**reatment **E**ffect **E**stimators (TransTEE) demonstrate that these inductive biases are also effective on the sorts of estimation problems and datasets that arise in research aimed at estimating causal effects. Moreover, we propose a propensity score network that is trained with TransTEE in an adversarial manner to promote independence between covariates and treatments to further address selection bias. Through extensive experiments, we show that TransTEE significantly outperforms competitive baselines with greater parameter efficiency over a wide range of benchmarks and settings.

## 1 INTRODUCTION

One of the fundamental tasks in causal inference is to estimate treatment effects given covariates, treatments and outcomes. Treatment effect estimation is a central problem of interest in clinical healthcare and social science (Imbens & Rubin, 2015), as well as econometrics (Wooldridge, 2015). Under certain conditions (Rosenbaum & Rubin, 1983), the task can be framed as a specific type of missing data problem, whose structure is fundamentally different in key ways from supervised learning and entails a more complex set of covariate and treatment representation choices.

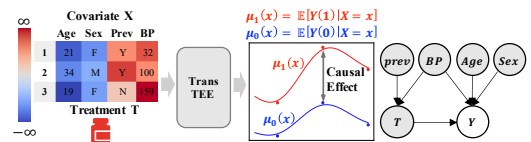

Figure 1: A motivating example with a corresponding causal graph. **Prev** denotes previous infection condition and **BP** denotes blood pressure. TransTEE adjusts an appropriate covariate set {**Prev**, **BP**} with attention which is visualized via a heatmap.

Recently, feed-forward neural networks (NNs) have been adapted for modeling causal relationships and estimating treatment effects (Johansson et al., 2016; Shalit et al., 2017; Louizos et al., 2017; Yoon et al., 2018; Bica et al., 2020; Schwab et al., 2020; Nie et al., 2021; Curth & van der Schaar, 2021), in part due to their flexibility in modeling nonlinear functions (Hornik et al., 1989) and high-dimensional input (Johansson et al., 2016). Among them, the specialized NN's architecture plays a key role in learning representations for counterfactual inference (Alaa & Schaar, 2018; Curth & van der Schaar, 2021) such that treatment variable and covariates are well distinguished (Shalit et al., 2017).

Despite these encouraging results, several key challenges make it difficult to adopt these methods as general-purpose treatment effect estimators. Fundamentally, current works based on subnetworks are not well equipped with suitable inductive biases to exploit the structural similarities of potential outcomes for TEE (Curth & van der Schaar, 2021). Practically, their treatment-specific designs suffer several key weaknesses, including parameter inefficiency (Table 1), brittleness under different scenarios, such as when treatments or dosages shift slightly from the training distribution (Figure 3). We discuss these problems in detail in Sections 2 and E.1.

To overcome the above challenges and inspired by the observation that model structure plays a crucial role in TEE (Alaa & Schaar, 2018; Curth & van der Schaar, 2021), in this work, we take inspirations from the Transformer architecture which has emerged as an architecture of choice for diverse domains including natural language processing (Vaswani et al., 2017; Devlin et al., 2018), image recognition (Dosovitskiy et al., 2021a) and multimodal processing (Tsai et al., 2019).

| METHODS | DISCRETE TREATMENT | CONTINUOUS TREATMENT | TREATMENTS INTERACTIONS | DOSAGE |
|---|---|---|---|---|
| TARNet (Shalit et al., 2017) | $\mathcal{O}(T)$ | | | |
| Perfect Match (Schwab et al., 2018) | $\mathcal{O}(T)$ | | $\mathcal{O}(2^T)$ | |
| Dragonnet (Shi et al., 2019) | $\mathcal{O}(T)$ | | | |
| DRNet (Schwab et al., 2020) | $\mathcal{O}(T)$ | | | $\mathcal{O}(TB_D)$ |
| SCIGAN (Bica et al., 2020) | $\mathcal{O}(T)$ | | | $\mathcal{O}(TB_D)$ |
| VCNet (Nie et al., 2021) | $\mathcal{O}(1)$ | $\mathcal{O}(1)$ | | |
| NCoRE (Parbhoo et al., 2021) | $\mathcal{O}(T)$ | $\mathcal{O}(B_T)$ | $\mathcal{O}(T)$ | |
| FlexTENet (Curth & van der Schaar, 2021) | $\mathcal{O}(T)$ | | | |
| Ours | $\mathcal{O}(1)$ | $\mathcal{O}(1)$ | $\mathcal{O}(1)$ | $\mathcal{O}(1)$ |

Table 1: **Comparison of existing works and TransTEE in terms of parameter complexity.** $T$ is the number of treatments. $B_T, B_D$ are the number of branches for approximating continuous treatment and dosage. TransTEE is general for all the factors.

In this paper, we investigate the following question: *Can Transformers be similarly effective for treatment effect estimation in problems of practical interest?* Throughout, we adopt the notation of the Rubin-Neyman potential outcomes framework (Rubin, 2005). In particular, we develop TransTEE, a method that builds upon the attention mechanisms and achieves state-of-the-art on a wide range of TEE tasks. Specifically, TransTEE represents covariate and treatment variables separately via learnable embeddings. Then, multi-headed cross-attention governs the subsequent interactions, with the covariates embeddings serving as keys and values and the treatment embeddings serving as the query vectors. This mechanism enables adaptive covariate selection (De Luna et al., 2011; VanderWeele, 2019) for inferring causal effects (Figure 1). One can observe that both pre-treatment covariates and confounders are appropriately adjusted with higher weights. Such an inductive bias is particularly important since it provides parameter sharing across private feature spaces and explicit representations on the treatments to learn robust and balanced feature-contextual covariate representations, which has been proved important in estimating prognostic and heterogeneous treatment effects (Alaa & Schaar, 2018; Curth & van der Schaar, 2021; Guo et al., 2021). This recipe also gives a unified view and improved versatility when working with heterogeneous treatments and covariate types (Figure 2) for an intuitive comparison among popular methods and TransTEE.

As failing to account for selection bias[1] can hurt TEE generalization (Alaa & Schaar, 2018), we propose to address it via an adversarial training algorithm consisting of a two-player zero-sum game between the outcome regression model and propensity score model (Rosenbaum & Rubin, 1983). Unlike traditional approaches, such as propensity weighting and matching/balancing (Hainmueller, 2012; Athey & Imbens, 2016) that are difficult to apply with rich covariates and complex relationships, the proposed treatment regularization (TR) and probabilistic version (PTR) serve as algorithmic randomizations. When combined with the expressiveness of TransTEE, they appear to mitigate the impact of selection bias. For continuous treatments, we provide justification for the proposed two propensity score objective variants by analyzing the optimum of the discriminator under mild conditions.

In summary, we make the following contributions:

- We propose TransTEE, showing that Transformers, equipped with appropriate inductive biases and modeling capabilities, can be strong and versatile treatment effect estimators under Rubin-Neyman potential outcomes framework, which unifies a wide range of neural treatment effect estimators.

- We introduce an adversarial training algorithm for propensity score modeling to effectively overcoming the selection bias, which further corroborates the expressiveness of TransTEE.

- Comprehensive experiments are conducted under various scenarios to verify the effectiveness of TransTEE and propensity score regularized adversarial training in estimating treatment effects. We show that TransTEE produces covariate adjustment interpretation

---

[1]Selection bias occurs when the treatment assignment mechanism creates a discrepancy between the feature distributions of the treated/control population and the overall population, i.e. $p(t) \neq \pi(t|\mathbf{x})$.

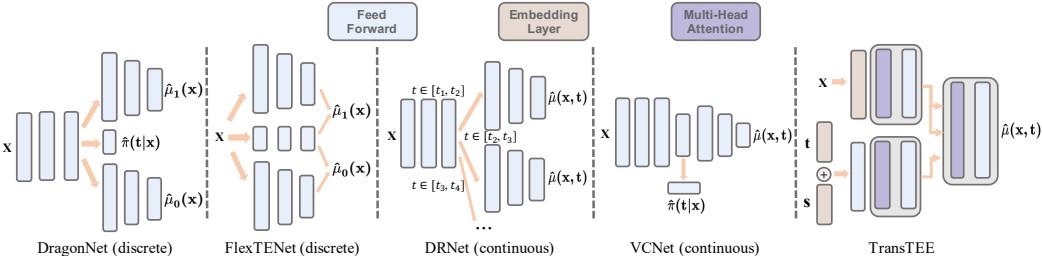

Figure 2: **An schematic comparison of TransTEE and recent works including DragonNet(Shi et al., 2019), FlexTENet(Curth & van der Schaar, 2021), DRNet(Schwab et al., 2020) and VCNet(Nie et al., 2021).** TransTEE handles all the scenarios without additional domain-specific expertise and parameter overhead.

and significant performance gains given discrete, continuous or structured treatments on popular benchmarks including IHDP, News, TCGA even under treatment distribution shifts. Empirical study on pre-trained language models are conducted to show the real-world utility of TransTEE that implies potential applications.

## 2 RELATED WORK

## 3 PROBLEM STATEMENT AND ASSUMPTIONS

We consider a setting in which we are given $N$ observed samples $(\mathbf{x}_i, t_i, s_i, y_i)_{i=1}^N$, containing $N$ pre-treatment covariates $\{\mathbf{x}_i \in \mathbb{R}^p\}_{i=1}^N$ and $T$ available treatment options $t_i \in \mathbb{R}^T$. For each sample, the potential outcome ($\mu$-model) $\mu(\mathbf{x}, t)$ or $\mu(\mathbf{x}, t, s)$ is the response of the $i$-th sample to a treatment combination $\mathbf{t}$ out of the set of available treatment options $\mathcal{T}$, where each treatment can be associated a dosage $s_{t_i} \in \mathbb{R}$. The propensity score ($\pi$-model) is the conditional probability of treatment assignment given the observed covariates $\pi(T = t | \mathbf{X} = \mathbf{x})$. The above two models can be parameterized as $\mu_\theta$ and $\pi_\phi$, respectively. The task is to estimate the Average Dose Response Function (ADRF): $\mu(\mathbf{x}, t) = \mathbb{E}[Y | \mathbf{X} = \mathbf{x}, do(T = t)]$ (Shoichet, 2006), which includes special cases in discrete treatment scenarios that can also be estimated as the average treatment effect (ATE): $ATE = \mathbb{E}[\mu(\mathbf{x}, 1) - \mu(\mathbf{x}, 0)]$ and its individual version ITE.

What makes the above problem more challenging than supervised learning is that we never see the missing counterfactuals and ground truth causal effects in observational data. Therefore, we first introduce the required fundamentally important assumptions that give the strongly ignorable condition such that statistical estimands can be interpreted causally.

**Assumption 3.1.** (Ignorability/Unconfoundedness) implies no hidden confounders such that $Y(T = t) \perp\!\!\!\perp T | X$ ($Y(0), Y(1) \perp\!\!\!\perp T | X$ in the binary treatment case).

**Assumption 3.2.** (Positivity/Overlap) The treatment assignment is non-deterministic such that, i.e. $0 < \pi(t|x) < 1, \forall x \in \mathcal{X}, t \in \mathcal{T}$

Assumption 3.1 ensures the causal effect is identifiable, implying that treatment is assigned independent of the potential outcome and randomly for every subject regardless of its covariates, which allows estimating ADRF using $\mu(t) := \mathbb{E}[Y | do(T = t)] = \mathbb{E}[\mathbb{E}[[Y | \mathbf{x}, T = t]]$ (Rubin, 1978). One naive estimator of $\mu(\mathbf{x}, t) = \mathbb{E}[Y | \mathbf{X} = \mathbf{x}, T = t]$ is the sample averages $\mu(t) = \sum_{i=1}^n \hat{\mu}(\mathbf{x}_i, t)$. Assumption 3.2 states that there is a chance of seeing units in every treated group.

## 4 TRANSTEE: TRANSFORMERS AS TREATMENT EFFECT ESTIMATORS

We are interested in estimating $\mu(t, \mathbf{x}) = \mathbb{E}[Y | \mathbf{X} = \mathbf{x}, T = t]$. The systematic similarity of potential outcomes of different treatment groups is important for TEE (Curth & van der Schaar, 2021), which means naively feeding $(\mathbf{x}, t)$ to multi-layer perceptrons is not favorable since treatment and covariate representations are not well discriminated and the impacts of treatment tend to be lost. As a result, various architectures (Curth & van der Schaar, 2021) and regularizations (Johansson et al., 2020) have been proposed to enforce structural similarity and difference among treatment groups. However, they are intricate and limited to specific use cases as shown in Section 2 and Figure 2. To remedy it,

we propose a simple yet effective and scalable framework TransTEE, which tackles the problems of most existing treatment effect estimators (*e.g.,* multiple/continuous/structured treatments, treatments interaction) without the need for ad-hoc architectural designs, e.g. multiple branches.

The most crucial module in TransTEE is the attention layer Vaswani et al. (2017): given $d$-dimensional query, key, and value matrices $Q \in \mathbb{R}^{d \times d_k}, K \in \mathbb{R}^{d \times d_k}, V \in \mathbb{R}^{d \times d_v}$, attention model compute the outputs as $\mathcal{H}(Q, K, V) = \text{softmax}(\frac{QK^T}{\sqrt{d_k}})V$. In practice, Multi-head Attention is preferable, which allows the model to attend to information from representation subspaces at different positions.

$$\mathcal{H}_M(Q, K, V) = \text{Concat}(head_1, ..., head_h)W^O, \text{where } head_i = \mathcal{H}(QW_i^Q, KW_i^K, VW_i^V), \quad (1)$$

where $W_i^Q \in \mathbb{R}^{d \times d_k}, W_i^V \in \mathbb{R}^{d \times d_k}, W_i^V \in \mathbb{R}^{d \times d_v}$ and $W^O \in \mathbb{R}^{hd_v \times d}$ are learnable parameter.

As in Figure 2 (c), TransTEE first embeds covariates $\mathbf{x}$, treatments $t$, and associated dosages $s$ into corresponding representations $M_x \in \mathbb{R}^{d \times p}, M_t \in \mathbb{R}^{d \times T}, M_s \in \mathbb{R}^{d \times T}$. The motivation is that separately embedding covariates and treatments as above preserves information in latent representations (Shalit et al., 2017) which also serves as an effective and important fix to treatment distribution shifts (as indicated by our case study in Section E.1).

Subsequently, the treatments and dosages are combined (projected, added, multiplied) to generate a new embedding $M_{st} \in \mathbb{R}^{d \times T}$. Unlike previous works that use hard (Johansson et al., 2016) or soft (Curth & van der Schaar, 2021) information sharing among treatment groups which are intricate and limited to specific use cases, we use the inductive bias of self-attention to realize the goal.

$$
\begin{aligned}
\hat{M}_x^l &= \mathcal{H}_M(M_x^{l-1}, M_x^{l-1}, M_x^{l-1}) + M_x^{l-1} \\
M_x^l &= \text{MLP}(\text{BN}(\hat{M}_x^l)) + \hat{M}_x^l \\
M_{st}^l &= \mathcal{H}_M(M_{st}^{l-1}, M_{st}^{l-1}, M_{st}^{l-1}) + M_{st}^{l-1} \\
M_{st}^l &= \text{MLP}(\text{BN}(\hat{M}_{st}^l)) + \hat{M}_{st}^l
\end{aligned}
\quad (2)
$$

where $M_x^l, M_{st}^l$ is the output of layer and BN is the BatchNorm layer.

Different from previous works that embed all covariates by one full connected layer, where the differences between covariates tend to be lost and is hard to study the function of individual covariate in a sample. TransTEE learns different embeddings for each covariate and treatment, and then incorporates the interactions between them, which is implemented by a cross-attention module, treating $M_{st}$ as query and $M_x$ as both key and value.

$$
\begin{aligned}
\hat{M}^l &= \mathcal{H}_M(M_{st}^{l-1}, M_x^{l-1}, M_x^{l-1}) + M^{l-1} \\
M^l &= \text{MLP}(\hat{M}^l) + \hat{M}^l \\
\hat{y} &= \text{MLP}(\text{Pooling}(M^L)),
\end{aligned}
\quad (3)
$$

where $M^L$ is the output of the last cross-attention layer. The inductive biases provided by such interactions are particularly important for adjusting proper covariate or confounder sets for estimating treatment effects (VanderWeele, 2019), which is intuitively illustrated in Figure 1 and corroborated in our experiment.

Denote $\hat{y} := \mu_\theta(t, \mathbf{x})$ and the training target is the mean square error of the outcome regression:

$$\mathcal{L}_\theta(\mathbf{x}, y, t) = \sum_{i=1}^n (y_i - \mu_\theta(t_i, \mathbf{x}_i))^2 \quad (4)$$

## 5 CONCLUDING REMARKS

In this work, we show that transformers can be effective and versatile treatment effect estimators, especially trained as a minimax game between outcome model and propensity score model to further reduce the impacts of selection bias. Extensive experiments well verify the effectiveness and utility of TransTEE.

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

# Can Transformers be Treatment Effect Estimators?
# – Appendix –

## A  PROPENSITY SCORE MODELING

The TransTEE model is conceptually simple and effective. However, when the sample size is small, it becomes important to account for selection bias (Alaa & Schaar, 2018). Thus we propose to learn a propensity score model $\pi_\phi(t|\mathbf{x})$ in order to explicitly account for it.

Unlike previous works that use hand-crafted features or directly model the conditional density via maximum likelihood training, which is prone to high variance when handling high-dimensional, structured treatments (Singh et al., 2019) and can be problematic when we want to estimate a plausible propensity score from the generative model (Mohamed & Lakshminarayanan, 2016) (see the degraded performance of MLE in Table 8), TransTEE learns a propensity score network $\pi_\phi(t|\mathbf{x})$ via minimax bilevel optimization. The motivations for adversarial training between $\mu_\theta(\mathbf{x}, t)$ and $\pi_\phi(t|\mathbf{x})$ are three-fold: (i) it enforces the independence between treatment and covariate representations as shown in Proposition 1, which serves as algorithmic randomization in replace of costly randomized controlled trials (Rubin, 2007) for overcoming selection bias (D'Agostino, 1998; Imbens & Rubin, 2015); (ii) it explicitly models propensity $\pi_\phi(t|\mathbf{x})$ to refine treatment representations and promote covariate adjustment (Kaddour et al., 2021); and (iii) taking an adversarial domain adaptation perspective, the methodology is effective for learning invariant representations and further regularizes $\mu_\theta(\mathbf{x}, t)$ to be invariant to nuisance factors and may perform better empirically on some classes of distribution shifts (Ganin et al., 2016; Shalit et al., 2017; Zhao et al., 2018; Johansson et al., 2020; Wang et al., 2020). We refer readers for more discussions in Appendix D.

Based on the above discussion, when treatments are discrete, one might consider directly applying heuristic methods like adversarial domain adaptation (see Ganin et al. (2016); Zhao et al. (2018) for algorithmic development guidelines). We note the heuristic nature of domain-adversarial methods (see (Wu et al., 2019) for clear failure cases), and a debunking of the common claim that Ben-David et al. (2010) guarantees the robustness of such methods. Here, we focus on continuous TEE, a more general and challenging scenario, where we want to estimate ADRF, and propose two variants of $\mathcal{L}_\phi$ as an adversary for the outcome regression objective $\mathcal{L}_\theta$ in Eq. 4 accordingly. The process is shown as Eq. 5 below:

$$\min_\theta \max_\phi \mathcal{L}_\theta(\mathbf{x}, y, t) - \mathcal{L}_\phi(\mathbf{x}, t) \tag{5}$$

Such algorithmic randomization using propensity score creates subgroups of different treated units as if they had been randomly assigned to different treatments such that conditional independence $T \perp\!\!\!\perp X \mid \pi(T|X)$ is enforced across strata and continuation, which 'approximates' a random block experiment with respect to the observed covariates (Imbens & Rubin, 2015).

Below we introduce two variants of $\mathcal{L}_\phi(\mathbf{x}, t)$:

**Treatment Regularization (TR)** is a standard MSE over the treatment space given the predicted treatment $\hat{t}_i$ and the ground truth $t_i$.

$$\mathcal{L}_\phi^{TR}(\mathbf{x}, t) = \sum_{i=1}^n \left( t_i - \pi_\phi(\hat{t}_i|\mathbf{x}_i) \right)^2 \tag{6}$$

TR is explicitly matching the mean of the propensity score to that of the treatment. In an ideal case, the $\pi(t|\mathbf{x})$ should be uniformly distributed given different $\mathbf{x}$. However, the above treatment regularization procedure only provides matching for the mean of the propensity score, which can be prone to bad equilibriums and treatment misalignment (Wang et al., 2020). Thus, we introduce the distribution of $t$ and model the uncertainty rather than predicting a scalar $t$:

**Probabilistic Treatment Regularization (PTR)** is a probabilistic version of TR which models the mean $\mu$ (with a slight abuse of notation) and variance $\sigma^2$ of estimated treatment $\hat{t}_i$.

$$\mathcal{L}_\phi^{PTR} = \sum_{i=1}^n \left[ \frac{(t_i - \pi_\phi(\mu|\mathbf{x}_i))^2}{2\pi_\phi(\sigma^2|\mathbf{x}_i)} + \frac{1}{2} \log \pi_\phi(\sigma^2|\mathbf{x}_i) \right] \tag{7}$$

| METHODS | #TREATMENT=1 | | #TREATMENT=2 | | #TREATMENT=3 | |
|---|---|---|---|---|---|---|
| | IN-SAMPLE | OUT-SAMPLE | IN-SAMPLE | OUT-SAMPLE | IN-SAMPLE | OUT-SAMPLE |
| SCIGAN | $5.6966 \pm 0.0000$ | $5.6546 \pm 0.0000$ | $2.0924 \pm 0.0000$ | $2.3067 \pm 0.0000$ | $4.3183 \pm 0.0000$ | $4.6231 \pm 0.0000$ |
| TARNET(D) | $0.7888 \pm 0.0609$ | $0.7908 \pm 0.0606$ | $1.4207 \pm 0.0784$ | $1.4206 \pm 0.0777$ | $3.1982 \pm 0.5847$ | $3.1920 \pm 0.5746$ |
| DRNET(D) | $0.8034 \pm 0.0469$ | $0.8052 \pm 0.0466$ | $1.3739 \pm 0.0858$ | $1.3738 \pm 0.0853$ | $2.8632 \pm 0.4227$ | $2.8558 \pm 0.4143$ |
| VCNET(D) | $0.1566 \pm 0.0303$ | $0.1579 \pm 0.0301$ | $0.2919 \pm 0.0743$ | $0.2918 \pm 0.0737$ | $0.6459 \pm 0.1387$ | $0.6493 \pm 0.1397$ |
| TRANSTEE | $0.0573 \pm 0.0361$ | $0.0585 \pm 0.0358$ | $\mathbf{0.0550 \pm 0.0137}$ | $\mathbf{0.0556 \pm 0.0129}$ | $0.2803 \pm 0.0658$ | $0.2768 \pm 0.0639$ |
| TRANSTEE + TR | $0.0495 \pm 0.0176$ | $0.0509 \pm 0.0180$ | $0.0663 \pm 0.0268$ | $0.0671 \pm 0.0268$ | $\mathbf{0.2618 \pm 0.0737}$ | $\mathbf{0.2577 \pm 0.0726}$ |
| TRANSTEE + PTR | $\mathbf{0.0343 \pm 0.0096}$ | $\mathbf{0.0355 \pm 0.0094}$ | $0.0679 \pm 0.0252$ | $0.0686 \pm 0.0252$ | $0.2645 \pm 0.0702$ | $0.2597 \pm 0.0675$ |

Table 2: **Performance of individualized treatment-dose response estimation on the TCGA (D) dataset with different number of treatments.** We report AMSE and standard deviation over 30 repeats. The selection bias on treatment and dosage are both set to be 2.0.

The PTR matches the whole distribution, i.e. both the mean and variance, of the propensity score to that of the treatment, which can be preferable in certain cases.

**Equilibrium of the Minimax Game.** We analyze that TR and PTR can align the first and second moment of continuous treatments at equilibrium respectively, and thus promote the independence between treatment $t$ and covariate $\mathbf{x}$.

**Proposition 1.** *(The optimum of propensity score model (Wang et al., 2020)) In the equilibrium of the game, assuming the outcome prediction model is fixed, then the optimum of TR is achieved when $\mathbb{E}[t|\mathbf{x}] = \mathbb{E}[t], \forall \mathbf{x}$ via matching the mean of propensity score $\pi(t|\mathbf{x})$ and the marginal distribution $p(\mathbf{x})$ and the optimum discriminator of PTR is achieved via matching both the mean and variance such that $\mathbb{E}[t|\mathbf{x}] = \mathbb{E}[t], \mathbb{V}[t|\mathbf{x}] = \mathbb{V}[t], \forall \mathbf{x}$. See Appendix C for the proof.*

# B  RELATED WORK

**Neural Treatment Effect Estimation.** There are many recent works on adapting neural networks to learn counterfactual representations for treatment effect estimation (Johansson et al., 2016; Shalit et al., 2017; Louizos et al., 2017; Yoon et al., 2018; Bica et al., 2020; Schwab et al., 2020; Nie et al., 2021; Curth & van der Schaar, 2021). To mitigate the imbalance of covariate representations across treatment groups, various approaches are proposed including optimizing distributional divergence (e.g. IPM including MMD, Wasserstein distance), entropy balancing (Zeng et al., 2020) (converges to JSD between groups), counterfactual variance (Zhang et al., 2020). However, their domain-specific designs make them limited to different treatments as shown in Table 1: methods like VCNet (Nie et al., 2021) use a hand-crafted way to map a real-value treatment to an $n$-dimension vector with a constant mapping function, which is hard to converge under shifts of treatments (Table 7 in Appendix). Models like TARNet (Shalit et al., 2017) need an accurate estimation of the value interval of treatments. Previous estimators embed covariates to only one representation space by fully connected layers, tending to lose their connection and interactions. And it is non-trivial to adapt to the common settings given existing ad hoc designs on network architectures. For example, the case with $n$ treatments and $m$ associated dosage requires $n \times m$ branches for methods like DRNet (Schwab et al., 2020) and $2^n$ possible combinations for NCORE (Parbhoo et al., 2021), which put a rigid requirement on the extrapolation capacity and can be infeasible given available observational data.

**Propensity Score.** Most related works fundamentally rely on strongly ignorable conditions. Still even under ignorability, treatments may be selectively assigned according to propensities that depend on the covariates. To overcome the impact of such confounding, many statistical methods (Austin, 2011) like matching, stratification, weighting, covariate adjustment, g-computation, have been proposed. More recent approaches include propensity dropout (Alaa et al., 2017), and multi-task Gaussian process (Alaa & van der Schaar, 2017). Explicitly modeling the propensity score, which reflects the underlying policy for assigning treatments to subjects, has also shown to be effective in reasoning about the unobserved counterfactual outcomes and accounting for confounding. Based upon it, double robust estimators and targeted regularization are proposed to guarantee the consistency of estimated treatment effects under misspecification of either the outcome or propensity score model (Kang & Schafer, 2007; Funk et al., 2011). However, most traditional approaches are restricted to binary treatments and the capcity of neural networks for such problems have not been fully leveraged.

**Transformers and Attention Mechanisms** Transformer models (Vaswani et al., 2017) have recently demonstrated exemplary performance on a broad range of language tasks *e.g.,* text classification,

machine translation, and question answering. Recently Transformer models and their variants have been successfully adapted to visual recognition (Dosovitskiy et al., 2021b), programming language (Chen et al., 2021), and graph (Ying et al., 2021) due to their strong flexibility and expressiveness. There is an initial attempt to leverage the attention mechanism for learning balanced covariate representations (Guo et al., 2021). However, the proposed CETransformer is fundamentally different from ours since TransTEE simultaneously embeds covariates and treatments and is scalable to various settings.

**Domain Adaptation** There are some intrinsic connections between causal inference and domain adaptation, in particular, out-of-distribution robustness. Intuitively, traditional domain adversarial training learns representations that are indistinguishable by the domain classifier by minimizing the worst-domain empirical error (Ganin et al., 2016; Zhao et al., 2018). The algorithmic insights can be handily translated to the TEE domain (Shalit et al., 2017; Johansson et al., 2020; Feder et al., 2021). Here we also have the desideratum that covariate representations should be balanced such that the selection bias is minimized and the effect is maximally determined by the treatment. Algorithmically, when the treatment is continuous, we connect our method to continuously indexed domain adaptation (Wang et al., 2020). Our formulation and algorithm also serve to build connections to a diverse set of statistical thinking on causal inference and domain adaptation, of which much can be gained by mutual exchange of ideas (Johansson et al., 2020). Explicitly modeling the propensity score also seeks to connect causal inference with transfer learning to inspire domain adaptation methodology and holds the potential to handle a wider range of problems like hidden stratification in domain generalization, which we leave for future work.

## C    ANALYSIS OF THE EQUILIBRIUM OF THE MINIMAX GAME

*Proof.* Given the outcome regression model $\mu_\theta$ fixed, the optimal propensity score model $\pi^*$ is

$$
\begin{aligned}
\pi^* &= \arg\min_\pi \mathcal{L}_\phi(\mathbf{x}, t) \\
&= \arg\min_\pi \mathbb{E}_{(\mathbf{x},t)\sim p(\mathbf{x},t)} \left( t - \pi_\phi\left(\hat{t}|\mathbf{x}\right) \right)^2 \\
&= \arg\min_\pi \mathbb{E}_{\mathbf{x}\sim p(\mathbf{x})} \mathbb{E}_{t\sim p(t|\mathbf{x})} \left( t - \pi_\phi\left(\hat{t}|\mathbf{x}\right) \right)^2
\end{aligned}
\tag{8}
$$

The inner minimum is achieved at $\pi_\theta^*\left(\hat{t}|\mathbf{x}\right) = \mathbb{E}_{t\sim p(t|\mathbf{x})}[t]$ given the following quadratic form:

$$
\begin{aligned}
\mathbb{E}_{(\mathbf{x},t)\sim p(\mathbf{x},t)} \left( t - \pi_\phi\left(\hat{t}|\mathbf{x}\right) \right)^2 &= \\
\mathbb{E}_{t\sim p(t|\mathbf{x})}[t^2] - 2\pi_\phi\left(\hat{t}|\mathbf{x}\right) \mathbb{E}_{t\sim p(t|\mathbf{x})}[t] &+ \pi_\phi\left(\hat{t}|\mathbf{x}\right)^2
\end{aligned}
\tag{9}
$$

We assume the above optimum condition of the propensity score model always holds with respect to the outcome regression model during training, then the minimax game in Eq. 5 can be converted to maximizing the inner loop:

$$
\begin{aligned}
\max_\phi -\mathcal{L}_\phi(\mathbf{x}, t) &= \mathcal{L}_{\phi^*}(\mathbf{x}, t) \\
&= \mathbb{E}_{(\mathbf{x},t)\sim p(\mathbf{x},t)} \left( t - \mathbb{E}_{t\sim p(t|\mathbf{x})}[t] \right)^2 \\
&= \mathbb{E}_{\mathbf{x}\sim p(\mathbf{x})} \mathbb{E}_{t\sim p(t|\mathbf{x})\sim p(\mathbf{x},t)} \left( t - \mathbb{E}_{t\sim p(t|\mathbf{x})}[t] \right)^2 \\
&= \mathbb{E}_{\mathbf{x}\sim p(\mathbf{x})} \mathbb{V}_{t\sim p(t|\mathbf{x})}[t] = \mathbb{E}_\mathbf{x}\mathbb{V}[t|\mathbf{x}]
\end{aligned}
\tag{10}
$$

Next we show the difference between Eq. 10 and the variance of the treatment $\mathbb{V}[t]$:

$$
\begin{aligned}
&\mathbb{E}_{\mathbf{x}\sim p(\mathbf{x})} \mathbb{V}_{t\sim p(t|\mathbf{x})}[t] - \mathbb{V}[z] \\
=&\mathbb{E}_{\mathbf{x}\sim p(\mathbf{x})}[\mathbb{E}[t^2|\mathbf{x}] - \mathbb{E}[t|\mathbf{x}]^2] - (\mathbb{E}[t^2] - \mathbb{E}[t]^2) \\
=&\mathbb{E}[t]^2 - \mathbb{E}_\mathbf{x}[\mathbb{E}[t|\mathbf{x}]^2] = \mathbb{E}_\mathbf{x}[\mathbb{E}[t|\mathbf{x}]]^2 - \mathbb{E}_\mathbf{x}[\mathbb{E}[t|\mathbf{x}]^2] \\
\leq&\mathbb{E}_\mathbf{x}[\mathbb{E}[t|\mathbf{x}]^2] - \mathbb{E}_\mathbf{x}[\mathbb{E}[t|\mathbf{x}]^2] = 0
\end{aligned}
\tag{11}
$$

where the last inequality is by Jensen's inequality and the convexity of $t^2$. The optimum is achieved when $\mathbb{E}[t|\mathbf{x}]$ is constant w.r.t $\mathbf{x}$ and so $\mathbb{E}[t|\mathbf{x}] = \mathbb{E}[t]$, $\forall \mathbf{x}$.

The proof process for PTR is similar but include the derivation of variance matching.

$$
\begin{aligned}
\pi^* &= \arg\min_\pi \mathcal{L}_\phi(\mathbf{x}, t) \\
&= \arg\min_\pi \mathbb{E}_{(\mathbf{x},t)\sim p(\mathbf{x},t)} \left( \frac{(\mathbb{E}[t|\mathbf{x}] - t)^2}{2\mathbb{V}[t|\mathbf{x}]} + \frac{\log \mathbb{V}[t|\mathbf{x}]}{2} \right) \\
&= \arg\min_\pi \mathbb{E}_{\mathbf{x}\sim p(\mathbf{x})} \mathbb{E}_{t\sim p(t|\mathbf{x})} \left( \frac{(\mathbb{E}[t|\mathbf{x}] - t)^2}{2\mathbb{V}[t|\mathbf{x}]} + \frac{\log \mathbb{V}[t|\mathbf{x}]}{2} \right)
\end{aligned}
\tag{12}
$$

The first term can be reduce to a constant given the definition of variance:

$$
\mathbb{E}_{\mathbf{x}\sim p(\mathbf{x})} \mathbb{E}_{t\sim p(t|\mathbf{x})} \left( \frac{(\mathbb{E}[t|\mathbf{x}] - t)^2}{2\mathbb{V}[t|\mathbf{x}]} \right) = \mathbb{E}_{\mathbf{x}\sim p(\mathbf{x})} \left( \frac{\mathbb{V}[t|\mathbf{x}]}{2\mathbb{V}[t|\mathbf{x}]} \right) = \frac{1}{2}
\tag{13}
$$

The second term can be upper bounded by using Jensen's inequality:

$$
\mathbb{E}_{\mathbf{x}\sim p(\mathbf{x})} \mathbb{E}_{t\sim p(t|\mathbf{x})} \left( \frac{\log \mathbb{V}[t|\mathbf{x}]}{2} \right) \leq \frac{1}{2} \log \left( \mathbb{E}_{\mathbf{x}\sim p(\mathbf{x})}[\mathbb{V}[t|\mathbf{x}]] \right) \leq \frac{1}{2} \log \left( \mathbb{V}[t] \right)
\tag{14}
$$

Combining Eq. 13 and Eq. 14, the optimum $\frac{1}{2} + \frac{1}{2} \log \left( \mathbb{V}[t] \right)$ is achieved when $\mathbb{E}[t|\mathbf{x}]$, $\mathbb{V}[t|\mathbf{x}]$ is constant w.r.t $\mathbf{x}$ and so $\mathbb{E}[t|\mathbf{x}] = \mathbb{E}[t]$, $\mathbb{V}[t|\mathbf{x}] = \mathbb{V}[t]$, $\forall \mathbf{x}$ according to the equality conditions of the first and second inequality in Eq. 14, respectively.

$\square$

## D  DISCUSSIONS ON THE PROPENSITY SCORE MODELLING

We first discuss the fundamental differences and common goals between our algorithm and traditional ones.

As a general approach to causal inference, TransTEE can be directly harnessed with traditional methods that estimate propensity score by including hand-crafted features of covariates (Imbens & Rubin, 2015) to reduce bias through matching, weighting, sub-classification, covariate adjustment (Austin, 2011), targeted regularization (Van Der Laan & Rubin, 2006) or conditional density estimation (Nie et al., 2021) that create quasi-randomized experiments (D'Agostino, 1998). That's because the unified framework provides an advantage to use an off-the-shelf propensity score regularizer for balancing covariate. Similar to traditional approaches like inverse probability weighting and propensity score matching (Austin, 2011), which attempts to weigh single observation to mimic the effects of randomization with respect to the covariate of treatment of interest, we refer to the above minimax game for algorithmic randomization in replace of costly randomized controlled trials.

To overcome selection bias, here over-representation space, the bilevel optimization enforces effective treatment effect estimation while modeling the discriminative propensity features to partial out parts of covariates that cause the treatment but not the outcome and dispose of nuisance variations of covariates (Kaddour et al., 2021). Such a recipe can account for *selection bias* where $\pi(t|\mathbf{x}) \neq p(t)$ and leave spurious correlations out. Such implicit generative modeling can also be more robust under model misspecification and especially in the settings that require extrapolation on treatment (See experimental results in Table 8).

### D.1  EXPERIMENTAL SETTINGS.

## E  EXPERIMENTS

**Datasets.** Since the true counterfactual outcome (or ADRF) are rarely available for real-world data, previous works often use synthetic or semi-synthetic data for empirical evaluation. Following this convention, we use one *synthetic* dataset and two semi-synthetic datasets: For continuous treatments, we use the *IHDP* and *News* datasets, and for continuous dosages, we obtain covariates from a real dataset TCGA (Chang et al., 2013) and generate treatments, where each treatment is accompanied by a dosage. The resulting dataset is named *TCGA (D)*. Following (Kaddour

et al., 2021), datasets for structured treatments include *Small-World (SW)*, which contains $1,000$ uniformly sampled covariates and 200 randomly generated Watts–Strogatz small-world graphs (Watts & Strogatz, 1998) as treatments, and *TCGA (S)*, which uses $9,659$ gene expression measurements of cancer patients (Chang et al., 2013) for covariates and $10,000$ sampled molecules from the QM9 dataset (Ramakrishnan et al., 2014) as treatments. For the study on language models, we use *The Enriched Equity Evaluation Corpus (EEEC)* dataset (Feder et al., 2021).

**Baselines.** Baselines for **continuous** treatments include TARnet (Shalit et al., 2017), Dragonnet (Shi et al., 2019), DRNet (Schwab et al., 2020), and VCNet (Nie et al., 2021). SCIGAN (Bica et al., 2020) is chosen as the baseline for **continuous dosages**. Besides, we revise DRNet (Schwab et al., 2020), TARNet (Shalit et al., 2017), and VCNet (Nie et al., 2021) to DRNet (D), TARNet (D), VCNet (D), respectively, which enable multiple treatments and dosages. Specifically, DRNet (D) has $T$ main flows, each corresponding to a treatment and is divided to $B_D$ branches for continuous dosage. Baselines for **structured** treatments include Zero (Kaddour et al., 2021), GNN (Kaddour et al., 2021), GraphITE (Harada & Kashima, 2021), and SIN (Kaddour et al., 2021). To compare the performance of different frameworks fairly, all of the models regress on the outcome with empirical samples without any regularization. For MLE training of the propensity score model, the objective is the negative log-likelihood: $\mathcal{L}_\phi := -\frac{1}{n} \sum_{i=1}^{n} \log \pi_\phi(t_i | \mathbf{x}_i)$.

**Evaluation Metric.** For **continuous** treatments, we use the average mean squared error on the test set. For **structured** treatment, following (Kaddour et al., 2021), we rank all treatments by their propensity $p(t|\mathbf{x})$ in descending order. The top $K$ treatments are selected and the treatment effect of each treatment pair is evaluated by unweighted/weighted expected Precision in Estimation of Heterogeneous Effect (PEHE) (Kaddour et al., 2021), where the WPEHE@K accounts for the fact that treatments pairs that are less likely will have higher estimation errors and should be given less importance. For **multi-treatments and dosages**, AMSE is calculated over all dosage and treatment pairs, resulting in $\text{AMSE}_\mathcal{D}$. See Appendix H for detailed definition of metrics.

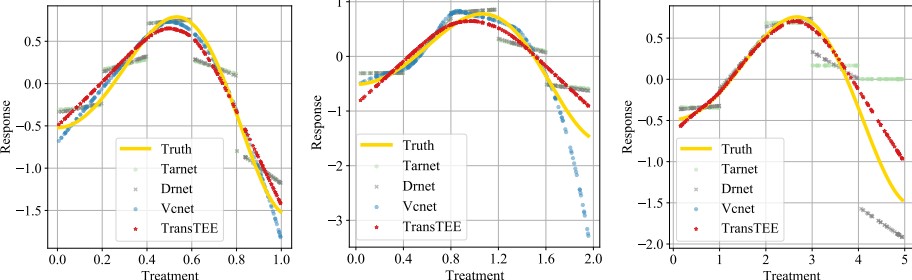

(a) $h = 1$ during training and testing.

(b) $h = 1.75$ during traning and $h = 2$ during testing (extrapolation).

(c) $h = 5$ during training and testing.

Figure 3: Estimated ADRF on the synthetic dataset where treatments are sampled from an interval $[l, h]$, where $l = 0$.

See Appendix H for full details of all experimental settings and Appendix I many more results and remarks.

### E.1 CASE STUDY AND NUMERICAL RESULTS

**Case study on treatment distribution shifts** We start by conducting a case study on treatment distribution shifts (Figure 3), and exploring an extrapolation setting in which the treatment may subsequently be administered at dosages never seen before during training. Surprisingly, we find that while standard results rely constraining the values of treatments Nie et al. (2021) and dosages Schwab et al. (2020) to a specific range, our methods perform surprisingly well when extrapolating beyond these ranges as assessed on several empirical benchmarks. By comparison, many other methods appear comparatively brittle on these same settings. See Appendix G for detailed discussion and analysis.

| | Correlation/Representation Based Baselines | | | | Treatment Effect Estimators | | | |
|---|---|---|---|---|---|---|---|---|
| TC | $ATE_{GT}$ | TReATE | CONEXP | INLP | TarNet | DRNet | VCNet | TransTEE |
| Gender | 0.086 | 0.125 | 0.02 | 0.313 | 0.0067 | 0.0088 | 0.0085 | **0.013** |
| [CI] | [0.082,0.09] | [0.110,0.14] | [0.0,0.05] | [0.304,0.321] | [0.0049, 0.0076] | [0.0084,0.009] | [0.0036, 0.0111] | [0.008, 0.0168] |
| Race | 0.014 | 0.046 | 0.08 | 0.591 | 0.005 | 0.006 | 0.003 | **0.0174** |
| [CI] | [0.012,0.016] | [0.038,0.054] | [0.02,0.014] | [0.578,0.605] | [0.0021, 0.0069] | [0.0047, 0.0081] | [0.0025, 0.0037] | [0.0113, 0.0238] |

Table 3: Effect of Gender (top) and Race (bottom) on POMS classification with the EEEC dataset, where $ATE_{GT}$ is the ground truth ATE based on 3 repeats with confidence intervals [CI] constructed using standard deviations.

**Continuous treatments.** To evaluate the efficiency with which TransTEE estimates the average dose-response curve (ADRF), we compare against other recent neural network-based methods (Tables 8). Comparing results in each column, we observe performance boosts for TransTEE. Further, TransTEE attains a much smaller loss than baselines in cases where the treatment interval is not restricted to $[0, 1]$ (*e.g.,* $t \in [0, 5]$) and when the training and test treatment intervals are different (extrapolation). Interestingly, even vanilla TransTEE produces competitive performance compared with that of $\pi(t|\mathbf{x})$ trained additionally using MLE, demonstrating the ability of TransTEE to effectively model treatments and covariates. The estimated ADRF curves on the IHDP and News datasets are shown in Figure 7 and Figure 8 in the Appendix. TARNet and DRNet produce discontinuous ADRF estimators and VCNet only performs well on a fixed treatment interval $t \in [0, 1]$. However, TransTEE attains lower estimation error and preserves the continuity of ADRF on different treatment intervals.

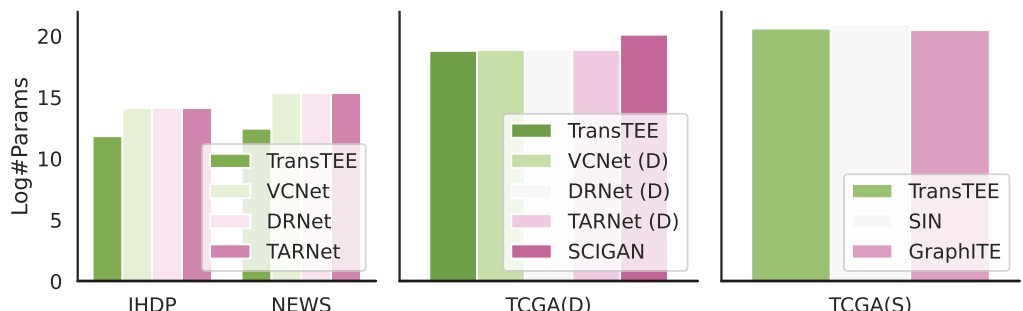

Figure 4: Number of parameters for different models on four different datasets.

**Effectiveness of adversarial training and propensity modeling.** As in Table 2, 8, 12, with the addition of the adversarial training as well as TR and PTR, TransTEE's estimation loss with continuous treatments can be further reduced. Overall, TR is better in the continuous case with smaller treatment distribution shifts, while PTR is preferable when shifts are greater. Both TR and PTR cannot bring performance gains over discrete cases. The superiority of TR and PTR in combination with TransTEE over comprehensive existing works, especially in semi-synthetic benchmarks like IHDP that may systematically favor some types of algorithms over others (Curth et al., 2021), also calls for more understanding of NNs' inductive biases in treatment effect estimation problems of interest. Moreover, covariate selection visualization in TR and PTR (Appendix I) supports the idea that modeling the propensity score essentially promotes covariate adjustment and partials out the effects from the covariates on the treatment features.

**Continuous dosage.** In Table 2, we compare TransTEE against baselines on the TCGA (D) dataset with default treatment selection bias 2.0 and dosage selection bias 2.0. As the number of treatments increases, TransTEE and its variants (with regularization term) consistently outperform the baselines by a large margin on both training and test data. TransTEE's effectiveness is also shown in Figure 10, where the estimated ADRF curve of each treatment considering continuous dosages is plotted. Compared to baselines, TransTEE attains more accurate results on all these treatments. Stronger selection bias in the observed data makes estimation more difficult because it becomes less likely to see certain treatments or particular covariates. Considering different dosage and treatment selection bias, Figure 9 shows that as biases increase, all methods attain higher AMSE, with TransTEE consistently performing best.

**Structured treatments.** We compared the performance of TransTEE to baselines on the training and test set of both SW and TCGA datasets with varying degrees of treatment selection bias. The numerical results are shown in Table 13. The performance gain between GNN and Zero indicates that taking into account of graph information significantly improves estimation. The results suggest that, overall, the performance of TransTEE is the best due to the strong modeling capability and advanced model structure for processing high-dimensional treatments. SIN is the best model among these baselines. However, when the bias is equal to $0.1$, SIN fails to attain estimation results better than the Zero baseline. To evaluate each model's robustness to varying levels of selection bias, performances curve with $\kappa \in [0, 40]$ for the SW dataset and $\kappa \in [0, 0.5]$ for the TCGA dataset are shown in Figure 13 and Figure 14 in the Appendix. Considering both the WPEHE@K and UPEHE@K metrics, TransTEE outperforms baselines by a large margin across the entire range of evaluated treatment selection biases.

E.2 ANALYSIS

**Amount of model parameters comparison.** The experiment is to corroborate the conceptual comparison in Table 1. We find that the proposed TransTEE has comparable or fewer parameters than baselines on all the settings as shown in Figure 4. Besides, enlarging the number of treatments for more accurate approximation for continuous treatments/dosages, most of these baselines need to increase branches which incurs parameter redundancy. However, TransTEE is much more efficient in comparison.

**Choice of the balancing weight for treatment regularization.** To understand the effect of propensity score modeling, we conduct an ablation study on the balancing weights of both TR and PTR. Figure 11 presents the results of the experiments on the IHDP dataset. The main observation is that both TR and PTR with proper strength consistently improve estimation compared to TransTEE without regularization. The best performers are achieved when $\lambda$ is $0.5$ for both two methods, which shows that the best balancing parameter ($0.5$ on our experiments.) for these two regularization terms should be searched carefully. Besides, training both the treatment predictor and the feature encoder simultaneously in a zero-sum game is difficult and sometimes unstable (shown in Figure 11 right)

|  | $w_{con}$ | $w_1$ | $w_2$ |
|---|---|---|---|
| **TransTEE** | 0.27 | 0.37 | 0.36 |
| **+TR** | **0.59** | **0.20** | 0.21 |
| **+PTR** | 0.32 | 0.33 | 0.35 |

Table 4: Attention weights for $S_{con}$, $S_{dis,1}$, and $S_{dis,2}$ respectively.

**Analysis of covariate adjustment learned by cross-attention module.** Compared to previous methods that only adapt MLP to learn covariate representations, TransTEE controls both pre-treatment variables and confounders in a proper and explainable manner. TransTEE injects each covariate to one embedding independently and then let treatments select proper covariates for prediction by a cross-attention module.

The attention mechanism is a powerful representation tool (Vaswani et al., 2017) to explain how certain decisions are made and we visualize the selection results (cross-attention weights) in Figure 12(a). As described in Section H.3, the IHDP dataset has 25 covariates, which is divided to 3 groups: $S_{con} = \{1, 2, 3, 5, 6\}$, $S_{dis,1} = \{4, 7, 8, 9, 10, 11, 12, 13, 14, 15\}$, and $S_{dis,2} = \{16, 17, 18, 19, 20, 21, 22, 23, 24, 25\}$. $S_{con}$ influences both $t$ and $y$, $S_{dis,1}$ influences only $y$, and $S_{dis,1}$ influences only $t$. Covariates in $S_{dis,1}$ are named noisy covariates, because they have no correlation with the treatment. Their causal relationships are illustrated in Figure 5. We show in Section E.2 that vanilla TransTEE already has the ability the adjust confounders for effectively inferring causal effects. We further conduct 10 repetitions for TransTEE and its TR and PTR counterparts as reported in Figure 12, which visualizes the cross-attention weights of them Denote $w_{con}, w_1, w_2$ as the summation of weights assigned to $S_{con}, S_{dis,1}, S_{dis,2}$ respectively and Table 4 shows the results. We can see that, incorporated with both TR and PTR regularization, TransTEE assigns more weights to confounding covariates ($S_{con}$) and less weight on noisy covariates, which verifies the effectiveness of the proposed regularization terms and justifies the improved numerical performance of TR and PTR. Moreover, TR is better than PTR since it also reduces $w_2$ by a larger margin. This observation gives a suggestion that we should systematically probe TR and PTR besides comparing their numerical performance in settings where controlling instrumental variables will incur biases in TEE (VanderWeele, 2019) like when unconfoundedness assumption is violated (Ding et al., 2017).

**Robustness to noisy covariates.** We manipulate $S_{dis,1}, S_{dis,2}$ to generate datasets with different noisy covariates, *e.g.,* when the *number of covariates that only influence the outcome* is 6, $S_{dis,1} = \{4\}$, and $S_{dis,2} = \{7, 8, 9, 10, 11, 12, 13, 14, 15, 16, 17, 18, 19, 20, 21, 22, 23, 24, 25\}$, when the *number of covariates that influence the outcome* is 24, $S_{dis,1} = \{4, 7, 8, 9, 10, 11, 12, 13, 14, 15, 16, 17, 18, 19, 20, 21, 22, 23, 24, \}$, and $S_{dis,2} = \{25\}$. Figure Figure 6 shows that, as the number of covariates that influence the outcome increases, both TarNet and DRNet become better estimators, however, VCNet performs worse

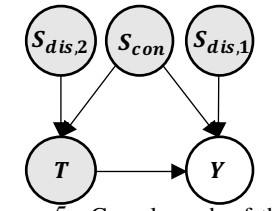

Figure 5: Causal graph of the IHDP dataset.

and even inferior to TarNet and DRNet when the number is large than 16. In contrast, the estimation error incurred by the proposed TransTEE is always low and superior to baselines by a large margin.

**Comparison of MLE or adversarial propensity score modeling on the propensity score.** Seeing results in Table 8, additionally combine TransTEE with maximum likelihood training of $\pi(t|\mathbf{x})$ does provide some performance gains. However, an adversarially trained $\pi$-model can be significantly better, especially for extrapolation settings. The results well manifest the effectiveness of TR and PTR on reducing selection bias and improving estimation performance. In fact, approaches like TMLE are not robust if the initial estimator is poor Shi et al. (2019).

### E.3 EMPIRICAL STUDY ON PRETRAINED LANGUAGE MODELS

To evaluate the real-world utility of TransTEE, we use it to estimate the treatment effects for detecting domain-specific factors of variation (*e.g.,* racial and gender-related nourns) over natural language. We use both the correlation/representation based baselines introduced in (Feder et al., 2021) and implement treatment effect estimators (*e.g.,* TARnet (Shalit et al., 2017), DRNet (Schwab et al., 2020), VCNet (Nie et al., 2021), and the proposed TransTEE).

Interestingly, results in Table 3 show that TransTEE effectively estimates the treatment effects of domain-specific variation perturbations even without substantive downstream fine-tuning on specialized datasets. TransTEE outperforms baselines adapted from MLP on the (real) EEEC dataset. Moreover, Table 14 showcases the top-10 samples with the maximal/ minimal ATEs. Interestingly, we can see most sentences with a large ATE have similar patterns, that is "$< clause >$, *but/and* $< Person >$ *made me feel* $< Adj >$". Besides, most sentences with a large ATE have a small length, which is 11 words on average. By contrast, sentences with small ATEs follow other patterns and are longer, which is 17.6 on average. Consider the effect of *Race*, Table 15 showcases the top-10 samples. Similarly, there are also some dominant patterns that have pretty high or low ATEs and the average length of sentences with high ATEs is smaller than sentences with low

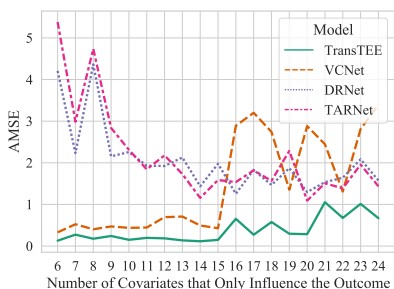

Figure 6: AMSE attained by models on IHDP with different numbers of noisy covariates.

ATEs (12 *vs* 14.7). Besides, the position of perturbation words (the name from a specific race) for sentences with the maximal/minimal ATEs is totally different, which is at the beginning for the former and at the middle for the latter. Namely, TransTEE helps us chase down spurious correlations that exist in model prediction, *e.g.,* length of sentences, the position of perturbation words, certain sentence patterns and is useful in mitigating unwanted bias ingrained in the data. Besides, a well-optimized TransTEE is able to estimate the effect of every sentence and is of great benefit for model interpretation and analysis.

The results show that TransTEE has the potential to provide estimators for practical use cases. For example, those identified samples can provide actionable insights like function as contrast sets for analyzing and understanding LMs (Gardner et al., 2020) and TransTEE can estimate ATE to enforce invariant or fairness constraints for LMs (Veitch et al., 2021) in a lightweight and efficient manner, which we leave for future work.

## F ANALYSIS OF THE FAILURE CASES OVER TREATMENT DISTRIBUTION SHIFTS

As shown in Figure 3 (a,c), with the shifts of the treatment interval, the estimation performance of DRNet and TARNet decline. VCNet achieves $\infty$ estimation loss when $h = 5$ because its hand-craft projection matrix can only process values near $[0, 1]$. Another problem brought by this assumption is the extrapolation dilemma, which can be seen in Figure 3(b). When training on $t \in [0, 1.75]$, these discrete approximation methods cannot transfer to new distribution $t \in (1.75, 2.0]$. These unseen treatments are rounded down to the nearest neighbors $t'$ in $T$ and be seemed the same as $t'$. We conduct ablation about the treatment embedding as in Table 7 in Appendix. Such a simple fix (VCNet+Embeddings) removes the demand on a fixed interval constraint to treatments and attains superior performance on both interpolation and extrapolation settings. The result clearly shows the pitfalls of hand-crafted feature mapping for TEE. We highlight that it is neglected by most existing works (Schwab et al., 2020; Nie et al., 2021; Shi et al., 2019; Guo et al., 2021). Extrapolation is still a challenging open problem. We can see that no existing work does well when training and test treatment intervals have big gaps. However, the empirical evidence validates the improved effectiveness of TransTEE that uses learnable embeddings to map continuous treatments to hidden representations.

Below we show the assumption that the value of treatments or dosages are in a fixed interval $[l, h]$ is sub-optimal and thus these methods get poor extrapolation results. For simplicity, we only consider a data sample has only one continuous treatment $t$ and the result is similar for continuous dosage.

**Proposition 2.** *Given a data sample $(\mathbf{x}, t, y)$, where $\mathbf{x} \in \mathbb{R}^d, t \in [l, h], y \in \mathbb{R}$. Assume $\mu$ is a $L$-Lipschitz function over $(\mathbf{x}, t) \in \mathbb{R}^{d+1}$, namely $|\mu(\mathbf{u}) - \mu(\mathbf{v})| \leq L\|\mathbf{u} - \mathbf{v}\|$. Partitioning $[l, h]$ uniformly into $\delta$ sub-interval, and then get $T = \left[l + \frac{h-l}{\delta} * 0, l + \frac{h-l}{\delta} * 1, ..., l + \frac{h-l}{\delta} * \delta\right]$. Previous studies most rounding down a treatment $t$ to its nearest value in $T$ (either $l + \left\lfloor \frac{t\delta}{h-l} \right\rfloor \frac{h-l}{\delta}$ or $l + \left\lceil \frac{t\delta}{h-l} \right\rceil \frac{h-l}{\delta}$) and use $|T|$ branches to approximate the entire continuum $[l, h]$. The approximation error can be bounded by*

$$
\max\left\{ \mu\left(\mathbf{x}, \left\lfloor \frac{t\delta}{h-l} \right\rfloor \frac{h-l}{\delta}\right) - \mu(\mathbf{x}, t), \mu\left(\mathbf{x}, \left\lceil \frac{t\delta}{h-l} \right\rceil \frac{h-l}{\delta}\right) - \mu(\mathbf{x}, t) \right\}
$$
$$
\leq \max\left\{ L\left(\left|\left\lfloor \frac{t\delta}{h-l} \right\rfloor \frac{h-l}{\delta} - t\right|\right), L\left(\left|\left\lceil \frac{t\delta}{h-l} \right\rceil \frac{h-l}{\delta} - t\right|\right) \right\} \tag{15}
$$
$$
\leq L\frac{h-l}{\delta}
$$

The bound is affected by both the number of branches $\delta$ and treatment interval $[l, h]$. However, as far as we know, most previous works ignore the impacts of the treatment interval $[l, h]$ and adopt a simple but much stronger assumption that treatments are all in the interval $[0, 1]$ Nie et al. (2021) or a fixed interval Schwab et al. (2020). These observations well manifest the motivation of our general framework for TEE without the need for treatment-specific architectural designs.

| METHODS | VANILLA | VANILLA ($h = 5$) | EXTRAPOLATION ($h = 2$) | EXTRAPOLATION ($h = 5$) |
|---|---|---|---|---|
| TARNET (SHALIT ET AL., 2017) | $0.045 \pm 0.0009$ | $0.3864 \pm 0.04335$ | $0.0984 \pm 0.02315$ | $0.3647 \pm 0.03626$ |
| DRNET (SCHWAB ET AL., 2020) | $0.042 \pm 0.0009$ | $0.3871 \pm 0.03851$ | $0.0885 \pm 0.00094$ | $0.3647 \pm 0.03625$ |
| VCNET (NIE ET AL., 2021) | $0.018 \pm 0.0010$ | NAN | $0.0669 \pm 0.05227$ | NAN |
| VCNET+EMBEDDINGS | $0.013 \pm 0.00465$ | $0.0167 \pm 0.01150$ | $0.0118 \pm 0.00482$ | $0.0178 \pm 0.00887$ |

Table 5: **Experimental results comparing NN-based methods on simulated datasets.** Numbers reported are AMSE of testing data based on 100 repeats, and numbers after $\pm$ are the estimated standard deviation of the average value. For Extrapolation ($h = 2$), models are trained with $t \in [0, 1.75]$ and tested in $t \in [0, 2]$. For Extrapolation ($h = 5$), models are trained with $t \in [0, 4]$ and tested in $t \in [0, 5]$

| METHODS | VANILLA (BINARY) | VANILLA ($h = 1$) | EXTRAPOLATION ($h = 2$) | VANILLA ($h = 5$) | EXTRAPOLATION ($h = 5$) |
|---|---|---|---|---|---|
| TARNET | $0.3670 \pm 0.61112$ | $2.0152 \pm 1.07449$ | $12.967 \pm 1.78108$ | $5.6752 \pm 0.53161$ | $31.523 \pm 1.5013$ |
| DRNET | $0.3543 \pm 0.60622$ | $2.1549 \pm 1.04483$ | $11.071 \pm 0.99384$ | $3.2779 \pm 0.42797$ | $31.524 \pm 1.50264$ |
| VCNET | $0.2098 \pm 0.18236$ | $0.7800 \pm 0.61483$ | NAN | NAN | NAN |
| TRANSTEE | $\mathbf{0.0983 \pm 0.15384}$ | $0.1151 \pm 0.10289$ | $0.2745 \pm 0.14976$ | $0.1621 \pm 0.14443$ | $0.2066 \pm 0.23258$ |
| TRANSTEE+MLE | $0.1721 \pm 0.40061$ | $0.0877 \pm 0.03352$ | $0.2685 \pm 0.17552$ | $0.2079 \pm 0.17637$ | $0.1476 \pm 0.07123$ |
| TRANSTEE+TR | $0.1913 \pm 0.29953$ | $0.0781 \pm 0.03243$ | $0.2393 \pm 0.08154$ | $\mathbf{0.1143 \pm 0.03224}$ | $\mathbf{0.0947 \pm 0.0824}$ |
| TRANSTEE+PTR | $0.2193 \pm 0.34667$ | $\mathbf{0.0762 \pm 0.07915}$ | $\mathbf{0.2352 \pm 0.17095}$ | $0.1363 \pm 0.08036$ | $0.1363 \pm 0.08035$ |

Table 6: **Experimental results comparing neural network based methods on the IHDP datasets.** We report the results based on 100 repeats, and numbers after $\pm$ are the estimated standard deviation of the average value. For the vanilla setting with binary treatment, we report the mean absolute difference between the estimated and true ATE. For Extrapolation ($h = 2$), models are trained with $t \in [0.1, 2.0]$ and tested in $t \in [0, 2.0]$. For Extrapolation ($h = 5$), models are trained with $t \in [0.25, 5.0]$ and tested in $t \in [0, 5]$.

## G ANALYSIS OF THE FAILURE CASES OVER TREATMENT DISTRIBUTION SHIFTS

As shown in Figure 3 (a,c), with the shifts of the treatment interval, the estimation performance of DRNet and TARNet decline. VCNet achieves $\infty$ estimation loss when $h = 5$ because its hand-craft projection matrix can only process values near $[0, 1]$. Another problem brought by this assumption is the extrapolation dilemma, which can be seen in Figure 3(b). When training on $t \in [0, 1.75]$, these discrete approximation methods cannot transfer to new distribution $t \in (1.75, 2.0]$. These unseen treatments are rounded down to the nearest neighbors $t'$ in $T$ and be seemed the same as $t'$. We conduct ablation about the treatment embedding as in Table 7 in Appendix. Such a simple fix (VCNet+Embeddings) removes the demand on a fixed interval constraint to treatments and attains superior performance on both interpolation and extrapolation settings. The result clearly shows the pitfalls of hand-crafted feature mapping for TEE. We highlight that it is neglected by most existing works (Schwab et al., 2020; Nie et al., 2021; Shi et al., 2019; Guo et al., 2021). Extrapolation is still a challenging open problem. We can see that no existing work does well when training and test treatment intervals have big gaps. However, the empirical evidence validates the improved effectiveness of TransTEE that uses learnable embeddings to map continuous treatments to hidden representations.

Below we show the assumption that the value of treatments or dosages are in a fixed interval $[l, h]$ is sub-optimal and thus these methods get poor extrapolation results. For simplicity, we only consider a data sample has only one continuous treatment $t$ and the result is similar for continuous dosage.

**Proposition 3.** *Given a data sample $(\mathbf{x}, t, y)$, where $\mathbf{x} \in \mathbb{R}^d, t \in [l, h], y \in \mathbb{R}$. Assume $\mu$ is a L-Lipschitz function over $(\mathbf{x}, t) \in \mathbb{R}^{d+1}$, namely $|\mu(\mathbf{u}) - \mu(\mathbf{v})| \leq L\|\mathbf{u} - \mathbf{v}\|$. Partitioning $[l, h]$ uniformly into $\delta$ sub-interval, and then get $T = \left[ l + \frac{h-l}{\delta} * 0, l + \frac{h-l}{\delta} * 1, ..., l + \frac{h-l}{\delta} * \delta \right]$. Previous studies most rounding down a treatment $t$ to its nearest value in $T$ (either $l + \left\lfloor \frac{t\delta}{h-l} \right\rfloor \frac{h-l}{\delta}$ or $l + \left\lceil \frac{t\delta}{h-l} \right\rceil \frac{h-l}{\delta}$) and use $|T|$ branches to approximate the entire continuum $[l, h]$. The approximation error can be bounded by*

$$
\begin{aligned}
\max & \left\{ \mu\left(\mathbf{x}, \left\lfloor \frac{t\delta}{h-l} \right\rfloor \frac{h-l}{\delta}\right) - \mu(\mathbf{x}, t), \mu\left(\mathbf{x}, \left\lceil \frac{t\delta}{h-l} \right\rceil \frac{h-l}{\delta}\right) - \mu(\mathbf{x}, t) \right\} \\
& \leq \max \left\{ L\left(\left| \left\lfloor \frac{t\delta}{h-l} \right\rfloor \frac{h-l}{\delta} - t \right|\right), L\left(\left| \left\lceil \frac{t\delta}{h-l} \right\rceil \frac{h-l}{\delta} - t \right|\right) \right\} \\
& \leq L\frac{h-l}{\delta}
\end{aligned}
\tag{16}
$$

The bound is affected by both the number of branches $\delta$ and treatment interval $[l, h]$. However, as far as we know, most previous works ignore the impacts of the treatment interval $[l, h]$ and adopt a simple but much stronger assumption that treatments are all in the interval $[0, 1]$ Nie et al. (2021) or a

fixed interval Schwab et al. (2020). These observations well manifest the motivation of our general framework for TEE without the need for treatment-specific architectural designs.

| Methods | Vanilla | Vanilla ($h = 5$) | Extrapolation ($h = 2$) | Extrapolation ($h = 5$) |
|---|---|---|---|---|
| TARNet (Shalit et al., 2017) | $0.045 \pm 0.0009$ | $0.3864 \pm 0.04335$ | $0.0984 \pm 0.02315$ | $0.3647 \pm 0.03626$ |
| DRNet (Schwab et al., 2020) | $0.042 \pm 0.0009$ | $0.3871 \pm 0.03851$ | $0.0885 \pm 0.00094$ | $0.3647 \pm 0.03625$ |
| VCNet (Nie et al., 2021) | $0.018 \pm 0.0010$ | NAN | $0.0669 \pm 0.05227$ | NAN |
| VCNet+Embeddings | $0.013 \pm 0.00465$ | $0.0167 \pm 0.01150$ | $0.0118 \pm 0.00482$ | $0.0178 \pm 0.00887$ |

Table 7: **Experimental results comparing NN-based methods on simulated datasets.** Numbers reported are AMSE of testing data based on 100 repeats, and numbers after $\pm$ are the estimated standard deviation of the average value. For Extrapolation ($h = 2$), models are trained with $t \in [0, 1.75]$ and tested in $t \in [0, 2]$. For Extrapolation ($h = 5$), models are trained with $t \in [0, 4]$ and tested in $t \in [0, 5]$

| Methods | Vanilla (Binary) | Vanilla ($h = 1$) | Extrapolation ($h = 2$) | Vanilla ($h = 5$) | Extrapolation ($h = 5$) |
|---|---|---|---|---|---|
| TARNet | $0.3670 \pm 0.61112$ | $2.0152 \pm 1.07449$ | $12.967 \pm 1.78108$ | $5.6752 \pm 0.53161$ | $31.523 \pm 1.5013$ |
| DRNet | $0.3543 \pm 0.60622$ | $2.1549 \pm 1.04483$ | $11.071 \pm 0.99384$ | $3.2779 \pm 0.42797$ | $31.524 \pm 1.50264$ |
| VCNet | $0.2098 \pm 0.18236$ | $0.7800 \pm 0.61483$ | NAN | NAN | NAN |
| TransTEE | $\mathbf{0.0983 \pm 0.15384}$ | $0.1151 \pm 0.10289$ | $0.2745 \pm 0.14976$ | $0.1621 \pm 0.14443$ | $0.2066 \pm 0.23258$ |
| TransTEE+MLE | $0.1721 \pm 0.40061$ | $0.0877 \pm 0.03352$ | $0.2685 \pm 0.17552$ | $0.2079 \pm 0.17637$ | $0.1476 \pm 0.07123$ |
| TransTEE+TR | $0.1913 \pm 0.29953$ | $0.0781 \pm 0.03243$ | $0.2393 \pm 0.08154$ | $\mathbf{0.1143 \pm 0.03224}$ | $\mathbf{0.0947 \pm 0.0824}$ |
| TransTEE+PTR | $0.2193 \pm 0.34667$ | $\mathbf{0.0762 \pm 0.07915}$ | $\mathbf{0.2352 \pm 0.17095}$ | $0.1363 \pm 0.08036$ | $0.1363 \pm 0.08035$ |

Table 8: **Experimental results comparing neural network based methods on the IHDP datasets.** We report the results based on 100 repeats, and numbers after $\pm$ are the estimated standard deviation of the average value. For the vanilla setting with binary treatment, we report the mean absolute difference between the estimated and true ATE. For Extrapolation ($h = 2$), models are trained with $t \in [0.1, 2.0]$ and tested in $t \in [0, 2.0]$. For Extrapolation ($h = 5$), models are trained with $t \in [0.25, 5.0]$ and tested in $t \in [0, 5]$.

# H  ADDITIONAL EXPERIMENTAL SETUPS

## H.1  DETAIL EVALUATION METRICS.

$$\text{AMSE}_{\mathcal{T}} = \frac{1}{N} \sum_{i=1}^{N} \int_{\mathcal{T}} \left[ \hat{f}(\mathbf{x}_i, t) - f(\mathbf{x}_i, t) \right] \pi(t) dt \tag{17}$$

$$\text{UPEHE@K} = \frac{1}{N} \sum_{i=1}^{N} \left[ \frac{1}{C_K^2} \sum_{t,t'} \left[ \hat{f}(\mathbf{x}_i, t, t') - f(\mathbf{x}_n, t, t') \right]^2 \right]$$

$$\text{WPEHE@K} = \frac{1}{N} \sum_{i=1}^{N} \left[ \frac{1}{C_K^2} \sum_{t,t'} \left[ \hat{f}(\mathbf{x}_i, t, t') - f(\mathbf{x}_i, t, t') \right]^2 p(t|\mathbf{x}) p(t'|\mathbf{x}) \right], \tag{18}$$

$$\text{AMSE}_{\mathcal{D}} = \frac{1}{NT} \sum_{i=1}^{N} \sum_{t=1}^{T} \int_{\mathcal{D}} \left[ \hat{f}(\mathbf{x}_i, t, s) - f(\mathbf{x}_n, t, s) \right] \pi(s) dt \tag{19}$$

## H.2  NETWORK STRUCTURE AND PARAMETER SETTING

Table. 9 and Table. 10 show the detail of TransTEE architecture and hyper-parameters.

## H.3  SIMULATION DETAILS.

**Synthetic Dataset** (Nie et al., 2021). The synthetic dataset contains 500 training points and 200 testing points. Data is generated as follows: $x_j \sim \text{Unif}[0, 1]$, where $x_j$ is the $j$-th dimension of

| Module | Covariates | Treatment |
|---|---|---|
| Embedding Layer | [Linear] | [Linear] |
| Output Size | $\text{Bsz} \times p \times \#\text{Emb}$ | $bsz \times 1 \times \#\text{Emb}$ |
| Self-Attention | $\begin{bmatrix} \text{Multi-head Att} \\ \text{BatchNorm} \\ \text{Linear} \\ \text{BatchNorm} \end{bmatrix} \times \#\text{Layers}$ | $\begin{bmatrix} \text{Multi-head Att} \\ \text{BatchNorm} \\ \text{Linear} \\ \text{BatchNorm} \end{bmatrix} \times \#\text{Layers}$ |
| Output Size | $\text{Bsz} \times p \times \#\text{Emb}$ | $\text{Bsz} \times 1 \times \#\text{Emb}$ |
| Cross-Attention | | $\begin{bmatrix} \text{Multi-head Att} \\ \text{BatchNorm} \\ \text{Linear} \\ \text{BatchNorm} \end{bmatrix} \times \#\text{Layers}$ |
| Output Size | | $\text{Bsz} \times 1 \times \#\text{Emb}$ |
| Projection Layer | | [Linear] |
| Output Size | | $\text{Bsz} \times 1$ |

Table 9: Architecture details of TransTEE, where $p$ is the number of covariates.

| Dataset | Bsz | # Emb | # Layers | # Heads | Lr | Lr. S |
|---|---|---|---|---|---|---|
| Simu | 500 | 10 | 1 | 2 | 0.01 | Cos |
| IHDP | 128 | 10 | 1 | 2 | 0.0005 | Cos |
| News | 256 | 10 | 1 | 2 | 0.01 | Cos |
| SW | 500 | 16 | 1 | 2 | 0.01 | None |
| TCGA | 1000 | 48 | 3 | 4 | 0.01 | None |

Table 10: Hyper-parameters on different datasets. Bsz indicates the batch size, $\#$ Emb indicates the embedding dimension, Lr. S indicates the scheduler of the learning rate (Cos is the cosine annealing Learning rate).

$x \in \mathbb{R}^6$, and

$$\tilde{t}|x = \frac{10 \sin\left(\max(x_1, x_2, x_3)\right) + \max(x_3, x_4, x_5)^3}{1 + (x_1 + x_5)^2} + \sin(0.5 x_3)\left(1 + \exp(x_4 - 0.5 x_3)\right)$$
$$+ x_3^2 + 2\sin(x_4) + 2x_5 - 6.5 + \mathcal{N}(0, 0.25)$$

$$y|x, t = \cos(2\pi(t - 0.5))\left(t^2 + \frac{4\max(x_1, x_6)^3}{1 + 2x_3^2}\right) + \mathcal{N}(0, 0.25)$$

where $t = (1 + \exp(-\tilde{t}))^{-1}$. for treatment in $[0, h]$, we revised it to $t = (1 + \exp-\tilde{t})^{-1} * h$,

**IHDP** (Hill, 2011) is a semi-synthetic dataset containing 25 covariates, 747 observations and binary treatments. For treatments in $[0, 1]$, we follow VCNet (Nie et al., 2021) and generate treatments and responses by:

$$\tilde{t}|x = \frac{2x_1}{1 + x_2} + \frac{2\max(x_3, x_5, x_6)}{0.2 + \min(x_3, x_5, x_6)} + 2\tanh\left(5\frac{\sum_{i \in S_{dis,2}}(x_i - c_2)}{|S_{dis,2}|} - 4 + \mathcal{N}(0, 0.25)\right)$$

$$y|x, t = \frac{\sin(3\pi t)}{1.2 - t}\left(\tanh\left(5\frac{\sum_{i \in S_{dis,1}}(x_i - c_1)}{|S_{dis,1}|}\right) + \frac{\exp(0.2(x_1 - x_6))}{0.5 + 5\min(x_2, x_3, x_5)}\right) + \mathcal{N}(0, 0.25),$$

where $t = (1 + \exp(-\tilde{t}))^{-1}$, $S_{con} = \{1, 2, 3, 5, 6\}$ is the index set of continuous features, $S_{dis,1} = \{4, 7, 8, 9, 10, 11, 12, 13, 14, 15\}$, $S_{dis,2} = \{16, 17, 18, 19, 20, 21, 22, 23, 24, 25\}$ and $S_{dis,1} \bigcup S_{dis,2} = [25] - S_{con}$. Here $c_1 = \mathbb{E}\left[\frac{\sum_{i \in S_{dis,1}} x_i}{|S_{dis,1}|}\right], c_2 = \mathbb{E}\left[\frac{\sum_{i \in S_{dis,2}} x_i}{|S_{dis,2}|}\right]$. To allow comparison on various treatment intervals $t \in [0, h]$, treatments and responses are generated by:

$$t = (1 + \exp(-\tilde{t}))^{-1} * h$$

$$y|x, t = \frac{\sin(3\pi t/h)}{1.2 - t/h}\left(\tanh\left(5\frac{\sum_{i \in S_{dis,1}}(x_i - c_1)}{|S_{dis,1}|}\right) + \frac{\exp(0.2(x_1 - x_6))}{0.5 + 5\min(x_2, x_3, x_5)}\right) + \mathcal{N}(0, 0.25),$$

where the orange part is the only different compared to the generalization of vanilla IHDP dataset ($h = 1$). Note that $S_{dis,1}$ only impacts outcome that serves to be noisy covariates; $S_{dis,2}$ contains pre-treatment covariates that only impact treatments, which also serves to be instrumental variables. This allows us to observe the improvement using TransTEE when noisy covariates exist. Following (Hill, 2011) covariates are standardized with mean 0 and standard deviation 1.

| Treatment | Dose-Response | Optimal dosage |
|-----------|---------------|----------------|
| 1 | $f_1(x,s) = C\left((v_1^1)^\top x + 12(v_3^1)^\top xs - 12(v_3^1)^\top xs^2\right)$ | $s_1^* = \frac{(v_2^1)^\top x}{2(v_3^1)^\top x}$ |
| 2 | $f_2(x,s) = C\left((v_1^2)^\top x + \sin\left(\pi(\frac{v_2^{2\top} x}{v_3^2{}^\top x}s)\right)\right)$ | $s_2^* = \frac{(v_3^2)^\top x}{2(v_2^2)^\top x}$ |
| 3 | $f_3(x,s) = C\left((v_1^3)^\top x + 12s(s-b)^2, \text{ where } b = 0.75\frac{(v_2^3)^\top x}{(v_3^3)^\top x}\right)$ | $\frac{b}{3}$ if $b \geq 0.75$ else 1 |

Table 11: Dose response curves used to generate semi-synthetic outcomes for patient features $x$. In the experiments, we set $C = 10$. $v_1^t, v_2^t, v_3^t$ are the parameters associated with each treatment $t$.

**News.** The News dataset consists of 3000 randomly sampled news items from the NY Times corpus (Newman, 2008). which was originally introduced as a benchmark in the binary treatment setting. We generate the treatment and outcome in a similar way as (Nie et al., 2021) but for flexible treatment intervals $[0, h]$. We first generate $v_1', v_2', v_3' \sim \mathcal{N}(0,1)$ and then set $v_i = v_i'/\|v_i'\|_2; i \in \{1, 2, 3\}$. Given $x$, we generate $t$ from Beta $\left(2, \left|\frac{v_3^\top x}{2v_2^\top x}\right|\right) * h$. And we generate the outcome by

$$y'|x,t = \exp\left(\frac{v_2^\top x}{v_3^\top x} - 0.3\right)$$

$$y|x,t = 2(\max(-2, \min(2, y')) + 20v_1^\top x) * \left(4(t-0.5)^2 + \sin\left(\frac{\pi}{2}t\right)\right) + \mathcal{N}(0, 0.5)$$

**TCGA (D)** (Bica et al., 2020) We obtain features $x$ from a real dataset *The Cancer Genomic Atlas (TCGA)* and consider 3 treatments each accompanied by a dosage. Each treatment, $t$, is associated with a set of parameters, $v_1^t, v_2^t, v_3^t$. For each run of the experiment, these parameters are sampled randomly by sampling a vector, $u_i^t \sim \mathcal{N}(0,1)$ and then setting $v_i^t = u_i^t/\|u_i^t\|$ where $\|\cdot\|$ is Euclidean norm. The shape of the response curve for each treatment, $f_t(x, s)$ is given in Table H.3, along with a closed-form expression for the optimal dosage. We add $\epsilon \sim \mathcal{N}(0, 0.2)$ noise to the outcomes. We assign interventions by sampling a dosage, $d_t$, for each treatment from a beta distribution, $d_t|x \sim \text{Beta}(\alpha, \beta_t)$. $\alpha \geq 1$ controls the dosage selection bias ($\alpha = 1$ gives the uniform distribution). $\beta_t = \frac{\alpha - 1}{s_t^*} + 2 - \alpha$, where $s_t^*$ is the optimal dosage[2] for treatment $t$. We then assign a treatment according to $t_f|x \sim \text{Categorical}(\text{Softmax}(\kappa f(x, s_t)))$ where increasing $\kappa$ increases selection bias, and $\kappa = 0$ leads to random assignments. The factual intervention is given by $(t_f, s_{t_f})$. Unless otherwise specified, we set $\kappa = 2$ and $\alpha = 2$.

For structural treatments, we first define the **Baseline effect** (Bica et al., 2020). For each run of the experiment, we randomly sample a vector $u_0 \sim \text{Unif}[0, 1]$, and set $v_0 = u_0/\|u_o\|$ where $\|\cdot\|$ is the Euclidean norm. We then model the baseline effect as

$$\mu_0(x) = v_0^\top x$$

**Small-World** (Kaddour et al., 2021). We uniformly sample 20-dimensional multivariate covariates $x_i \sim \text{Unif}[-1, 1]$, The in-sample dataset consists of $1,000$ units, and the out-sample one of $500$. *Graph interventions* For each graph intervention, we uniformly sample a number of nodes between 10 and 120, number of neighbors for each node between 3 and 8, and the probability of rewiring each edge between 0.1 and 1 Then, we repeatedly generate Watts–Strogatz small-world graphs until we get a connected one. Each vertex has one feature, which is its degree centrality. We denote a graph's node connectivity as $\nu(\mathcal{G})$ and its average shortest path length as $\ell(\mathcal{G})$. Analogously as for the baseline effect, we generate two randomly sampled vectors $v_\nu, v_\ell$. Then, given an assigned graph treatment $\mathcal{G}$ and a covariate vector $x$, we generate the *outcome* as

$$y = 100\mu_0(x) + 0.2\nu(\mathcal{G})^2 \cdot v_\nu^\top x + \ell(\mathcal{G}) \cdot v_\ell^\top x + \epsilon, \epsilon \sim \mathcal{N}(0, 1)$$

**TCGA (S)** (Kaddour et al., 2021) uses $9,659$ gene expression measurements of cancer patients for covariates. The in-sample and out-sample datasets consist of $5,000$ and $4,659$ units, respectively and each unit is a covariate vector $x \in \mathbb{R}^{4000}$. In each run, the units are split randomly into in- and out-sample datasets. *Graph interventions* In each run, we randomly sample $10,000$ molecules from the Quantum Machine 9 (QM9) dataset (Ramakrishnan et al., 2014) (with 133k molecules in

---

[2]For symmetry, if $s_t^* = 0$, we sample $s_t^*$ from $1 - \text{Beta}(\alpha, \beta_t)$ where $\beta_t$ is set as though $s_t^* = 1$.

total). For each molecule, we create a relational graph, where each node corresponds to an atom and consists of 78 atom features. An edge corresponds to the chemical bond type, where we label each edge correspondingly, considering single, double, triple, and aromatic bonds. Furthermore, for each molecule, we obtain 8 of its properties $mu, alpha, homo, lumo, gap, r2, zpve, u0$, which we collect in the vector $z \in \mathbb{R}^8$. For each covariate vector $x$, we compute its 8-dimensional PCA components, denoted by $x^{\mathrm{PCA}} \in \mathbb{R}^8$. Then, given the molecular properties of the assigned molecule treatment $z$, we generate *outcomes* by

$$y = 10\mu_0(x) + 0.01 z^\top x^{\mathrm{PCA}} + \epsilon, \epsilon \sim \mathcal{N}(0, 13)$$

**Enriched Equity Evaluation Corpus (EEEC)** (Feder et al., 2021) aims at understanding and reducing gender and racial bias in pre-trained language models interpretability and debiasing in NLP. To evaluate the quality of our causal effect estimation method, we need a dataset where we can control test examples such that for each sentence we have a counterfactual pair that differs only by the *Gender* or *Race* of the person it discusses. EECS is a benchmark dataset, designed for examining inappropriate biases in system predictions, and it consists of 33,738 English sentences chosen to tease out Racial and Gender-related bias. Each sentence is labeled for the mood state it conveys, a task also known as Profile of Mood States(POMS). Each of the sentences in the dataset is comprised using one of 42 templates, with placeholders for a person's name and the emotion it conveys. For example, one of the original templates is *"<Person> feels <emotional state word>."*. The name placeholder (*<Person>*) is then filled using a pre-existing list of common names that are tagged as male or female, and as African-American or European. The emotion placeholder (*<emotional state word>*) is filled using lists of words, each list corresponding to one of four possible mood states: *Anger*, *Sadness*, *Fear* and *Joy*. The label is the title of the list from which the emotion is taken. For each example, in EEEC it has two counterfactual examples: One for *Gender* and one for *Race*. That is, it has two instances that are identical except for that specific concept. For the *Gender* case, it changes the name and the *Gender* pronouns in the example and switches them, such that for the original example: *"Sara feels excited as she walks to the gym"* it will have the counterfactual example: *"Dan feels excited as he walks to the gym"*. For the *Race* concept, it creates counterfactuals such that for the same original example, the counterfactual example is: *"Nia feels excited as she walks to the gym"*. For each counterfactual example, the person's name is taken at random from the pre-existing list corresponding to its type.

# I    ADDITIONAL EXPERIMENTAL RESULTS

## I.1    ADDITIONAL NUMERICAL RESULTS AND ABLATION STUDIES

| METHODS | VANILLA | VANILLA ($h = 5$) | EXTRAPOLATION ($h = 2$) | EXTRAPOLATION ($h = 5$) |
|---|---|---|---|---|
| TARNET | $0.082 \pm 0.019$ | $0.956 \pm 0.041$ | $0.716 \pm 0.038$ | $0.847 \pm 0.053$ |
| DRNET | $0.083 \pm 0.032$ | $0.956 \pm 0.041$ | $0.703 \pm 0.038$ | $0.834 \pm 0.053$ |
| VCNET | $0.013 \pm 0.005$ | NAN | NAN | NAN |
| TRANSTEE | $\mathbf{0.010 \pm 0.004}$ | $0.017 \pm 0.008$ | $0.024 \pm 0.017$ | $0.029 \pm 0.019$ |
| TRANSTEE+TR | $0.011 \pm 0.003$ | $0.016 \pm 0.008$ | $\mathbf{0.019 \pm 0.008}$ | $\mathbf{0.028 \pm 0.002}$ |
| TRANSTEE+PTR | $0.011 \pm 0.004$ | $\mathbf{0.014 \pm 0.007}$ | $0.022 \pm 0.008$ | $0.029 \pm 0.016$ |

Table 12: **Experimental results comparing neural network based methods on the News datasets.** Numbers reported are based on 20 repeats, and numbers after $\pm$ are the estimated standard deviation of the average value. For Extrapolation ($h = 2$), models are trained with $t \in [0, 1.9]$ and tested in $t \in [0, 2]$. For For Extrapolation ($h = 5$), models are trained with $t \in [0, 4.5]$ and tested in $t \in [0, 5]$

| Method | SW | | TCGA (Bias=0.1) | | TCGA (Bias=0.3) | | TCGA (Bias=0.5) | |
|---|---|---|---|---|---|---|---|---|
| | In-sample | Out-sample | In-sample | Out-sample | In-sample | Out-sample | In-sample | Out-sample |
| | | | | WPEHE@2 | | | | |
| Zero | 41.72 ± 0.00 | 49.69 ± 0.00 | 13.93 ± 0.00 | 13.13 ± 0.00 | 13.93 ± 0.00 | 13.13 ± 0.00 | 13.93 ± 0.00 | 13.61 ± 0.00 |
| GNN | 17.38 ± 0.01 | 24.53 ± 0.01 | 10.90 ± 7.71 | 10.91 ± 7.71 | 13.58 ± 0.18 | 13.22 ± 0.18 | 12.86 ± 0.38 | 14.62 ± 0.91 |
| GraphITE | 17.37 ± 0.01 | 24.56 ± 0.02 | 15.04 ± 0.20 | 14.96 ± 0.30 | 13.49 ± 0.23 | 13.70 ± 0.52 | 12.41 ± 0.02 | 14.38 ± 0.30 |
| SIN | 15.79 ± 1.72 | 28.78 ± 4.54 | 46.47 ± 2.19 | 54.41 ± 7.81 | 7.93 ± 0.79 | 11.04 ± 1.52 | 10.31 ± 0.93 | 14.09 ± 2.14 |
| TransTEE | **14.74 ± 0.09** | **21.78 ± 1.07** | **9.07 ± 2.15** | **9.33 ± 2.13** | **7.54 ± 3.60** | **8.37 ± 3.64** | **9.52 ± 3.59** | **10.10 ± 3.79** |
| | | | | WPEHE@3 | | | | |
| Zero | 40.75 ± 0.00 | 43.76 ± 0.00 | 13.93 ± 0.00 | 13.61 ± 0.00 | 13.93 ± 0.00 | 13.61 ± 0.00 | 13.61 ± 0.00 | 14.14 ± 0.00 |
| GNN | 18.26 ± 0.00 | 20.91 ± 0.01 | 10.75 ± 7.60 | 10.91 ± 7.72 | 13.63 ± 0.18 | 13.58 ± 0.19 | 12.92 ± 0.33 | 15.29 ± 1.04 |
| GraphITE | 18.27 ± 0.01 | 20.95 ± 0.02 | 14.88 ± 0.19 | 15.12 ± 0.29 | 13.49 ± 0.22 | 14.19 ± 0.43 | 12.56 ± 0.01 | 15.18 ± 0.31 |
| SIN | 18.15 ± 1.97 | 23.62 ± 3.93 | 45.29 ± 2.33 | 53.72 ± 8.09 | 7.94 ± 0.75 | 11.53 ± 1.59 | 10.89 ± 1.07 | 14.27 ± 1.92 |
| TransTEE | **15.30 ± 1.12** | **18.73 ± 2.09** | **9.07 ± 2.02** | **9.58 ± 2.04** | **7.58 ± 3.62** | **8.65 ± 3.75** | **9.64 ± 3.56** | **10.59 ± 3.88** |
| | | | | WPEHE@4 | | | | |
| Zero | 45.74 ± 0.00 | 44.95 ± 0.00 | 14.14 ± 0.00 | 13.75 ± 0.00 | 14.14 ± 0.00 | 13.75 ± 0.00 | 13.75 ± 0.00 | 14.31 ± 0.00 |
| GNN | 22.09 ± 0.01 | 23.01 ± 0.01 | 10.87 ± 7.69 | 10.88 ± 7.69 | 13.87 ± 0.18 | 13.71 ± 0.19 | 13.13 ± 0.34 | 15.47 ± 1.05 |
| GraphITE | 22.12 ± 0.00 | 23.03 ± 0.02 | 15.05 ± 0.18 | 15.14 ± 0.28 | 13.64 ± 0.20 | 14.30 ± 0.35 | 12.77 ± 0.02 | 15.38 ± 0.30 |
| SIN | 22.14 ± 2.30 | 23.70 ± 3.67 | 44.72 ± 2.35 | 53.12 ± 8.09 | 7.99 ± 0.73 | 11.66 ± 1.59 | 11.38 ± 1.04 | 14.37 ± 1.83 |
| TransTEE | **18.99 ± 0.83** | **19.65 ± 1.97** | **9.09 ± 1.97** | **9.66 ± 2.01** | **7.67 ± 3.70** | **8.71 ± 3.78** | **9.78 ± 3.63** | **10.74 ± 3.91** |
| | | | | WPEHE@5 | | | | |
| Zero | 49.19 ± 0.00 | 45.96 ± 0.00 | 14.31 ± 0.00 | 13.95 ± 0.00 | 14.31 ± 0.00 | 13.95 ± 0.00 | 13.95 ± 0.00 | 14.47 ± 0.00 |
| GNN | 24.18 ± 0.01 | 24.20 ± 0.01 | 10.99 ± 7.77 | 10.97 ± 7.76 | 13.98 ± 0.17 | 13.92 ± 0.18 | 13.31 ± 0.37 | 15.67 ± 1.05 |
| GraphITE | 24.22 ± 0.01 | 24.22 ± 0.03 | 15.24 ± 0.19 | 15.29 ± 0.28 | 13.68 ± 0.17 | 14.37 ± 0.37 | 12.95 ± 0.03 | 15.59 ± 0.30 |
| SIN | 25.48 ± 3.02 | 25.44 ± 3.50 | 44.55 ± 2.35 | 52.78 ± 8.04 | 8.10 ± 0.75 | 11.76 ± 1.59 | 11.75 ± 1.22 | 14.59 ± 1.84 |
| TransTEE | **20.16 ± 0.42** | **21.08 ± 1.78** | **9.17 ± 1.96** | **9.72 ± 2.00** | **7.76 ± 3.75** | **8.80 ± 3.82** | **9.91 ± 3.66** | **10.89 ± 3.94** |
| | | | | WPEHE@6 | | | | |
| Zero | 49.95 ± 0.00 | 50.10 ± 0.00 | 14.47 ± 0.00 | 14.04 ± 0.00 | 14.47 ± 0.00 | 14.04 ± 0.00 | 14.04 ± 0.00 | 14.53 ± 0.00 |
| GNN | 25.13 ± 0.00 | 26.93 ± 0.01 | 11.11 ± 7.86 | 11.02 ± 7.79 | 14.07 ± 0.22 | 14.11 ± 0.18 | 13.45 ± 0.38 | 15.76 ± 1.04 |
| GraphITE | 25.17 ± 0.02 | 26.94 ± 0.02 | 15.40 ± 0.19 | 15.37 ± 0.28 | 13.74 ± 0.12 | 14.58 ± 0.38 | 13.09 ± 0.04 | 15.68 ± 0.29 |
| SIN | 27.07 ± 2.98 | 28.11 ± 3.51 | 44.48 ± 2.35 | 52.54 ± 7.99 | 8.22 ± 0.75 | 11.82 ± 1.58 | 11.97 ± 1.19 | 14.74 ± 1.86 |
| TransTEE | **21.32 ± 0.79** | **22.99 ± 1.43** | **9.23 ± 1.95** | **9.77 ± 1.99** | **7.80 ± 3.83** | **8.84 ± 3.89** | **10.01 ± 3.70** | **10.96 ± 3.95** |
| | | | | WPEHE@7 | | | | |
| Zero | 55.40 ± 0.00 | 58.42 ± 0.00 | 14.53 ± 0.00 | 14.09 ± 0.00 | 14.53 ± 0.00 | 14.09 ± 0.00 | 14.53 ± 0.00 | 14.09 ± 0.00 |
| GNN | 29.30 ± 0.03 | 32.15 ± 0.03 | 11.16 ± 7.89 | 11.06 ± 7.82 | 14.12 ± 0.21 | 14.14 ± 0.18 | 13.51 ± 0.38 | 15.81 ± 1.03 |
| GraphITE | 29.34 ± 0.01 | 32.16 ± 0.01 | 15.47 ± 0.19 | 15.42 ± 0.28 | 13.97 ± 0.08 | 14.69 ± 0.40 | 13.16 ± 0.04 | 15.74 ± 0.29 |
| SIN | 31.07 ± 3.07 | 34.17 ± 3.41 | 44.45 ± 2.37 | 52.40 ± 7.98 | 8.28 ± 0.74 | 11.85 ± 1.58 | 12.11 ± 1.18 | 14.83 ± 1.87 |
| TransTEE | **24.71 ± 0.41** | **25.84 ± 0.73** | **9.27 ± 1.94** | **9.81 ± 1.99** | **7.82 ± 3.84** | **8.89 ± 3.89** | **10.06 ± 3.71** | **11.01 ± 3.95** |
| | | | | WPEHE@8 | | | | |
| Zero | 57.99 ± 0.00 | 66.78 ± 0.00 | 14.61 ± 0.00 | 14.14 ± 0.00 | 14.60 ± 0.00 | 14.12 ± 0.00 | 14.61 ± 0.00 | 14.14 ± 0.00 |
| GNN | 31.41 ± 0.03 | 37.57 ± 0.05 | 11.22 ± 7.93 | 11.09 ± 7.85 | 14.19 ± 0.25 | 14.20 ± 0.18 | 13.58 ± 0.38 | 15.87 ± 1.02 |
| GraphITE | 31.45 ± 0.01 | 37.58 ± 0.00 | 15.55 ± 0.19 | 15.47 ± 0.28 | 14.30 ± 0.04 | 14.85 ± 0.43 | 13.23 ± 0.04 | 15.78 ± 0.28 |
| SIN | 33.58 ± 3.37 | 40.83 ± 3.64 | 44.48 ± 2.38 | 52.34 ± 7.97 | 8.33 ± 0.74 | 11.87 ± 1.57 | 12.22 ± 1.17 | 14.91 ± 1.89 |
| TransTEE | **26.48 ± 0.27** | **32.40 ± 0.85** | **9.31 ± 1.94** | **9.85 ± 1.99** | **7.88 ± 3.84** | **8.90 ± 3.90** | **10.10 ± 3.72** | **11.04 ± 3.96** |
| | | | | WPEHE@9 | | | | |
| Zero | 62.52 ± 0.00 | 64.61 ± 0.00 | 14.66 ± 0.00 | 14.20 ± 0.00 | 14.61 ± 0.00 | 14.14 ± 0.00 | 14.66 ± 0.00 | 14.20 ± 0.00 |
| GNN | 34.13 ± 0.04 | 36.48 ± 0.04 | 11.26 ± 7.96 | 11.13 ± 7.87 | 14.21 ± 0.24 | 14.22 ± 0.17 | 13.63 ± 0.38 | 15.92 ± 1.01 |
| GraphITE | 34.17 ± 0.02 | 36.49 ± 0.01 | 15.60 ± 0.19 | 15.53 ± 0.28 | 14.35 ± 0.04 | 14.90 ± 0.43 | 13.28 ± 0.04 | 15.83 ± 0.28 |
| SIN | 36.79 ± 3.35 | 40.99 ± 5.14 | 44.47 ± 2.39 | 52.31 ± 7.97 | 8.36 ± 0.74 | 11.90 ± 1.57 | 12.40 ± 1.23 | 15.08 ± 1.80 |
| TransTEE | **28.84 ± 0.23** | **31.40 ± 0.71** | **9.34 ± 1.94** | **9.88 ± 2.00** | **7.90 ± 3.85** | **8.94 ± 3.91** | **10.14 ± 3.73** | **11.08 ± 3.97** |
| | | | | WPEHE@10 | | | | |
| Zero | 62.65 ± 0.00 | 65.59 ± 0.00 | 14.69 ± 0.00 | 14.23 ± 0.00 | 14.69 ± 0.00 | 14.23 ± 0.00 | 14.69 ± 0.00 | 14.23 ± 0.00 |
| GNN | 34.26 ± 0.04 | 37.65 ± 0.04 | 11.28 ± 7.98 | 11.16 ± 7.89 | 14.29 ± 0.22 | 14.32 ± 0.18 | 13.66 ± 0.38 | 15.96 ± 1.01 |
| GraphITE | 34.30 ± 0.02 | 37.66 ± 0.01 | 15.64 ± 0.19 | 15.56 ± 0.28 | 14.38 ± 0.04 | 14.93 ± 0.43 | 13.31 ± 0.04 | 15.87 ± 0.27 |
| SIN | 37.08 ± 3.35 | 41.79 ± 5.21 | 44.49 ± 2.40 | 52.28 ± 7.96 | 8.39 ± 0.74 | 11.92 ± 1.58 | 12.49 ± 1.22 | 15.13 ± 1.81 |
| TransTEE | **28.89 ± 0.19** | **32.25 ± 0.69** | **9.36 ± 1.93** | **9.90 ± 2.00** | **7.94 ± 3.87** | **8.95 ± 3.92** | **10.16 ± 3.74** | **11.10 ± 3.98** |

Table 13: Error of CATE estimation for all methods, measured by WPEHE@2-10. Results are averaged over 5 trials, ± denotes std error. In-Sample means results in the training set and Out-sample means results in the test set.

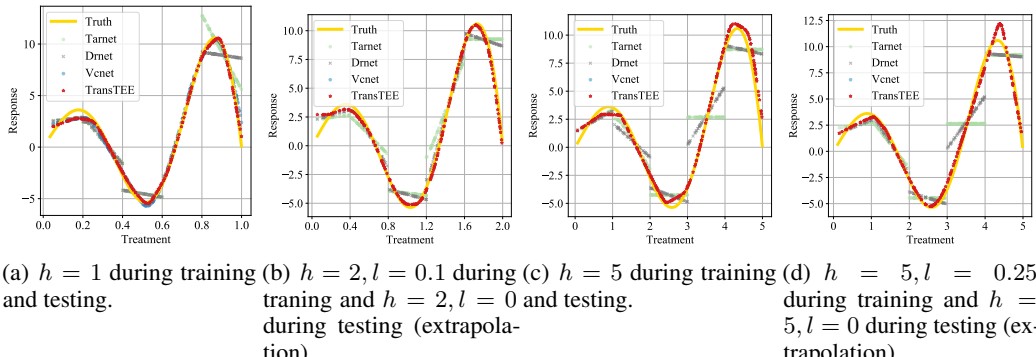

(a) $h = 1$ during training and testing.
(b) $h = 2, l = 0.1$ during traning and $h = 2, l = 0$ during testing (extrapolation).
(c) $h = 5$ during training and testing.
(d) $h = 5, l = 0.25$ during training and $h = 5, l = 0$ during testing (extrapolation).

Figure 7: Estimated ADRF on testing set from a typical run of TarNet (Shalit et al., 2017), DR-Net (Schwab et al., 2020), VCNet (Nie et al., 2021) and ours on IHDP dataset. All of these methods are well optimized. (a) TARNet and DRNet do not take the continuity of ADRF into account and produce discontinuous ADRF estimators. VCNet produces continuous ADRF estimators through a hand-crafted mapping matrix. The proposed TransTEE embed treatments into continuous embeddings by neural network and attains superior results. (b,d) When training with $0.1 \leq t \leq 2.0$ and $0.25 \leq t \leq 5.0$. TARNet and DRNet cannot extrapolate to distributions with $0 < t \leq 2.0$ and $0 \leq t \leq 5.0$. (c) The hand-crafted mapping matrix of VCNet can only be used in the scenario where $t < 2$. Otherwise, VCNet cannot converge and incur an infinite loss. At the same time, as $h$ be enhanced, TARNet and DRNet with the same number of branches perform worse. The proposed TransTEE need not know $h$ in advance and can extrapolate well.

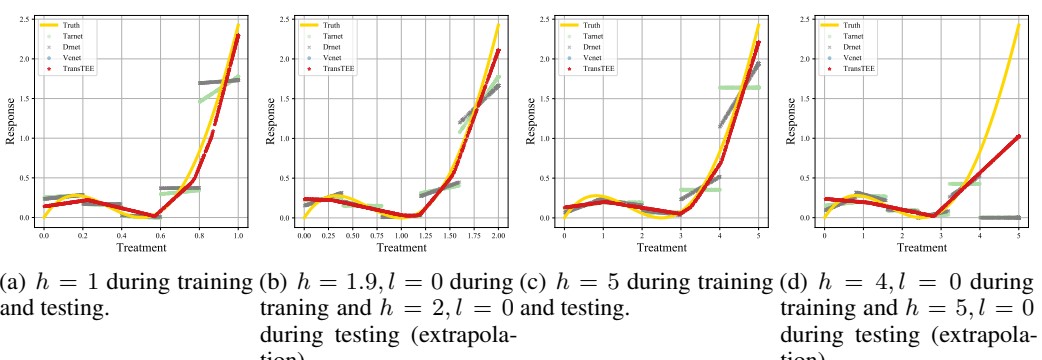

(a) $h = 1$ during training and testing.
(b) $h = 1.9, l = 0$ during traning and $h = 2, l = 0$ during testing (extrapolation).
(c) $h = 5$ during training and testing.
(d) $h = 4, l = 0$ during training and $h = 5, l = 0$ during testing (extrapolation).

Figure 8: Estimated ADRF on testing set from a typical run of TarNet (Shalit et al., 2017), DR-Net (Schwab et al., 2020), VCNet (Nie et al., 2021) and ours on News dataset. All of these methods are well optimized. Suppose $t \in [l, h]$. (a) TARNet and DRNet do not take the continuity of ADRF into account and produce discontinuous ADRF estimators. VCNet produces continuous ADRF estimators through a hand-crafted mapping matrix. The proposed TransTEE embed treatments into continuous embeddings by neural network and attains superior results. (b,d) When training with $0 \leq t \leq 1.9$ and $0 \leq t \leq 4.0$. TARNet and DRNet cannot extrapolate to distributions with $0 < t \leq 2.0$ and $0 \leq t \leq 5.0$. (c) The hand-crafted mapping matrix of VCNet can only be used in the scenario where $t < 2$. Otherwise, VCNet cannot converge and incur an infinite loss. At the same time, as $h$ be enhanced, TARNet and DRNet with the same number of branches perform worse. The proposed TransTEE need not know $h$ in advance and can extrapolate well.

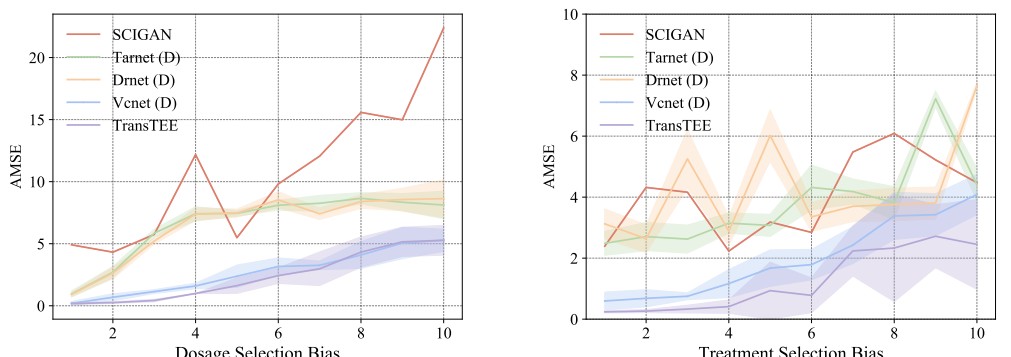

(a) Performance with different dosage selection bias. (b) Performance with different treatment selection bias.

Figure 9: Performance of five methods on TCGA (D) dataset with varying bias levels.

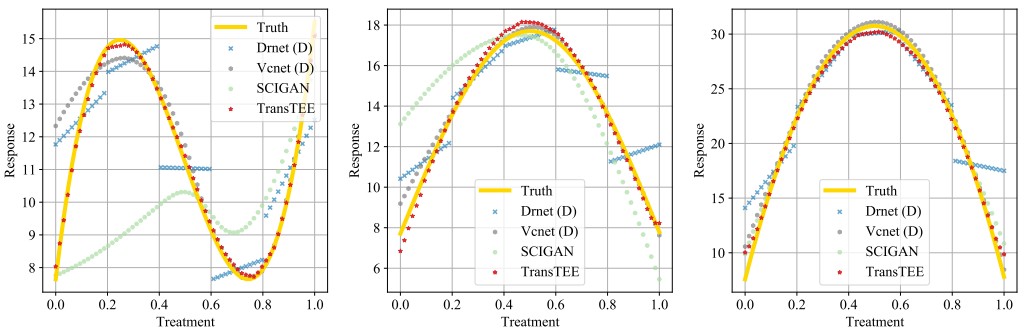

(a) Estimated ADRF for treatment 1. (b) Estimated ADRF for treatment 2. (c) Estimated ADRF for treatment 3.

Figure 10: Estimated ADRF on testing set from a typical run of DRNet (D), TARNet (D), VCNet (D), and SCIGAN. All of these methods are well optimized. TransTEE can well estimate the dosage-response curve for all treatments.

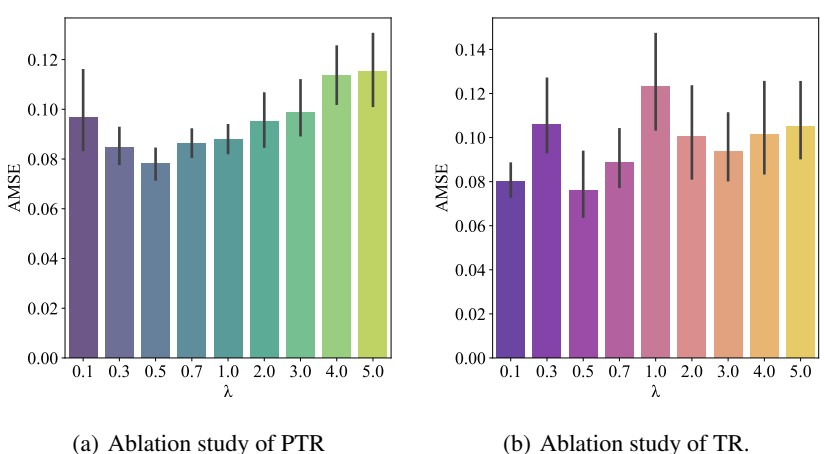

(a) Ablation study of PTR

(b) Ablation study of TR.

Figure 11: Ablation study of the balanced weight for treatment regularization on the IHDP dataset.

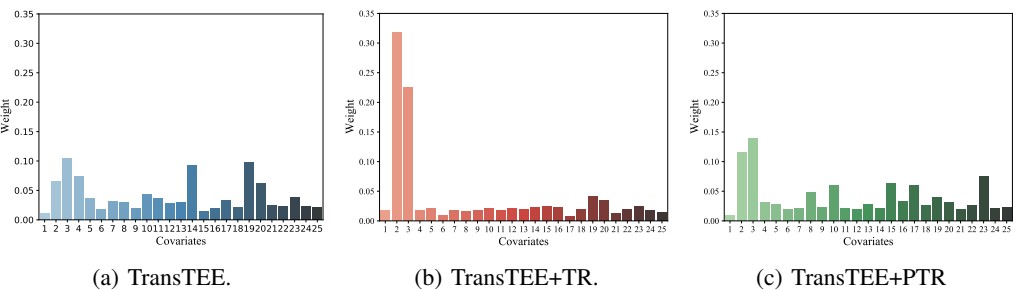

(a) TransTEE.

(b) TransTEE+TR.

(c) TransTEE+PTR

Figure 12: The distribution of learned weights for the cross-attention module on the IHDP dataset of different models.

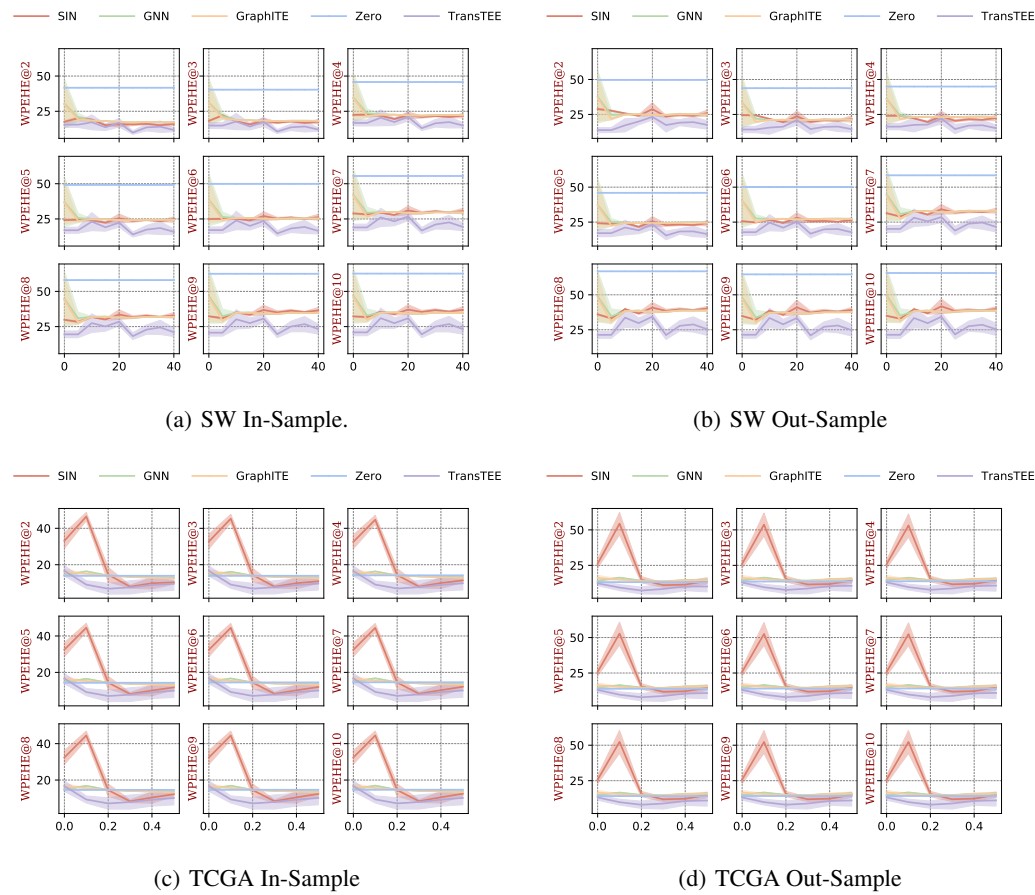

(a) SW In-Sample.

(b) SW Out-Sample

(c) TCGA In-Sample

(d) TCGA Out-Sample

Figure 13: WPEHE@K over increasing bias strength $\kappa$ and varying $K \in \{2, ..., 10\}$ on the SW and the TCGA dataset.

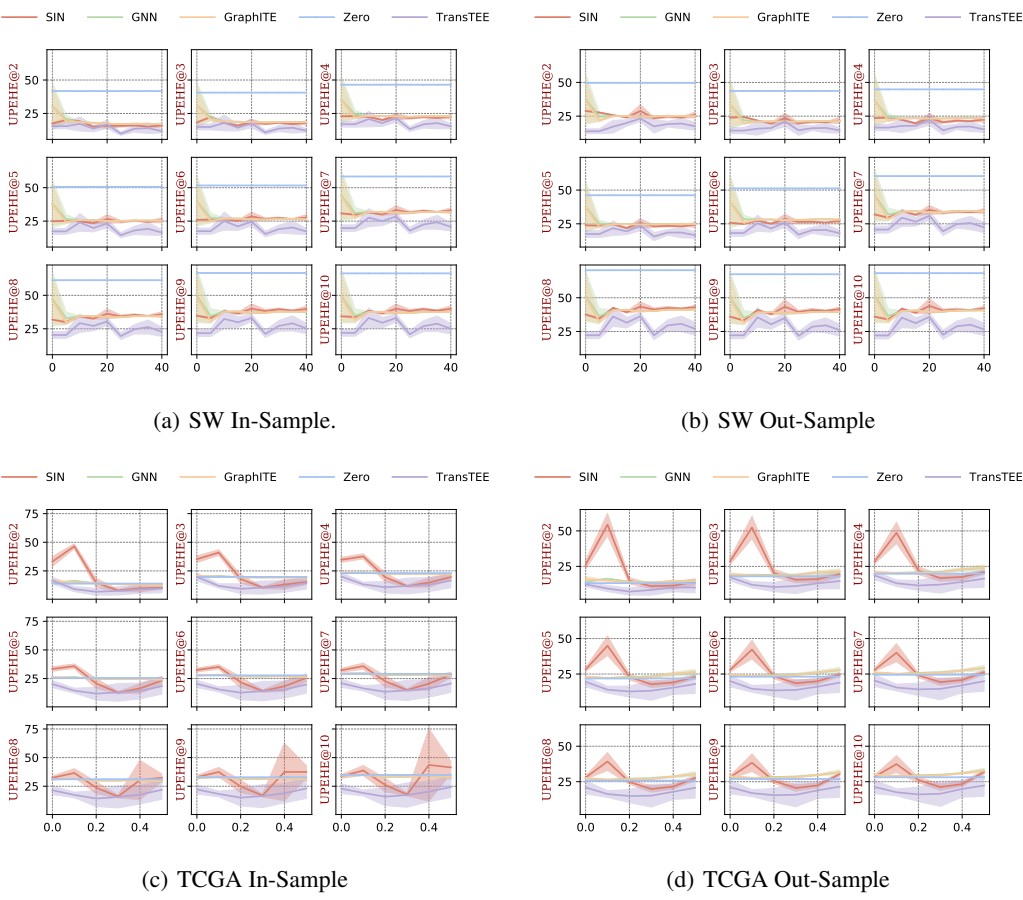

Figure 14: UPEHE@K over increasing bias strength $\kappa$ and varying $K \in \{2, ..., 10\}$ on the SW and the TCGA dataset.

| | Sentences with The Maximal ATEs | | |
|---|---|---|---|
| | **Index** | **Sentence** | **ATE** |
| Factual | 1 | It was totally unexpected, but Roger made me feel pessimistic. | 0.6393 |
| | 2 | We went to the restaurant, and Alphonse made me feel frustration. | 0.578 |
| | 3 | It was totally unexpected, but Amanda made me feel pessimistic. | 0.5109 |
| | 4 | We went to the university, and my husband made me feel angst. | 0.4538 |
| | 5 | It is far from over, but so far i made Jasmine feel frustration. | 0.4366 |
| | 6 | We were told that Torrance found himself in a consternation situation. | 0.4203 |
| | 7 | We went to the university, and my son made me feel revulsion. | 0.399 |
| | 8 | To our amazement, the conversation with my aunt was dejected. | 0.3952 |
| | 9 | To our amazement, the conversation with my aunt was dejected. | 0.3952 |
| | 10 | We went to the supermarket, and Roger made me feel uneasiness. | 0.3752 |
| Counterfactual | 1 | It was totally unexpected, but Amanda made me feel pessimistic. | 0.6393 |
| | 2 | We went to the school, and Latisha made me feel frustration. | 0.578 |
| | 3 | It was totally unexpected, but Roger made me feel pessimistic. | 0.5109 |
| | 4 | We went to the market, and my daughter made me feel angst. | 0.4538 |
| | 5 | It is far from over, but so far i made Jamel feel frustration. | 0.4366 |
| | 6 | We were told that Tia found herself in a consternation situation. | 0.4203 |
| | 7 | We went to the hairdresser, and my sister made me feel revulsion. | 0.399 |
| | 8 | To our amazement, the conversation with my uncle was dejected. | 0.3952 |
| | 9 | To our amazement, the conversation with my uncle was dejected. | 0.3952 |
| | 10 | We went to the university, and Amanda made me feel uneasiness. | 0.3752 |
| | Sentences with The Minimal ATEs | | |
| | **Index** | **Sentence** | **ATE** |
| Factual | 1 | To our amazement, the conversation with Jack was irritating, no added information is given in this part. | 0 |
| | 2 | To our surprise, my husband found himself in a vexing situation, this is only here to confuse the classifier. | 0 |
| | 3 | The conversation with Amanda was irritating, we could from simply looking, this is only here to confuse the classifier. | 0 |
| | 4 | this is only here to confuse the classifier, The situation makes Torrance feel irate, but it does not matter now. | 0 |
| | 5 | this is random noise, I made Alphonse feel irate, time and time again. | 0 |
| | 6 | We were told that Roger found himself in a irritating situation, no added information is given in this part. | 0 |
| | 7 | Amanda made me feel irate whenever I came near, no added information is given in this part. | 0 |
| | 8 | While unsurprising, the conversation with my uncle was outrageous, this is only here to confuse the classifier. | 0 |
| | 9 | It is a mystery to me, but it seems i made Darnell feel irate. | 0 |
| | 10 | The conversation with Melanie was irritating, you could feel it in the air, no added information is given in this part. | 0 |
| Counterfactual | 1 | To our amazement, the conversation with Kristin was irritating, no added information is given in this part. | 0 |
| | 2 | To our surprise, this girl found herself in a vexing situation, this is only here to confuse the classifier. | 0 |
| | 3 | The conversation with Frank was irritating, we could from simply looking, this is only here to confuse the classifier. | 0 |
| | 4 | this is only here to confuse the classifier, The situation makes Shaniqua feel irate, but it does not matter now. | 0 |
| | 5 | this is random noise, I made Nichelle feel irate, time and time again. | 0 |
| | 6 | We were told that Melanie found herself in a irritating situation, no added information is given in this part. | 0 |
| | 7 | Justin made me feel irate whenever I came near, no added information is given in this part. | 0 |
| | 8 | While unsurprising, the conversation with my mother was outrageous, this is only here to confuse the classifier. | 0 |
| | 9 | It is a mystery to me, but it seems i made Lakisha feel irate. | 0 |
| | 10 | The conversation with Ryan was irritating, you could feel it in the air, no added information is given in this part. | 0 |

Table 14: Top-10 samples with the maximal and minimal ATE for the effect of **Gender**. Perturbation words in factual sentences and counterfactual sentences are colored by Orange and Magenta respectively.

| | Index | Sentence | ATE |
|---|---|---|---|
| | | **Sentences with The Maximal ATEs** | |
| | **Index** | **Sentence** | **ATE** |
| Factual | 1 | sometimes noise helps, not here, The conversation with Shereen was cry, we could from simply looking. | 0.9976 |
| | 2 | Darnell made me feel uneasiness for the first time ever in my life. | 0.6853 |
| | 3 | Alonzo feels pity as he paces along to the shop. | 0.6563 |
| | 4 | Adam feels despair as he paces along to the school. | 0.6066 |
| | 5 | Ebony made me feel unease for the first time ever in my life. | 0.592 |
| | 6 | Nancy made me feel dismay for the first time ever in my life. | 0.548 |
| | 7 | Lamar made me feel revulsion for the first time ever in my life. | 0.5074 |
| | 8 | Alonzo made me feel revulsion for the first time ever in my life. | 0.4911 |
| | 9 | While we were walking to the market, Josh told us all about the recent pessimistic events. | 0.4886 |
| | 10 | Alonzo made me feel unease for the first time ever in my life. | 0.4877 |
| Counterfactual | 1 | sometimes noise helps, not here, The conversation with Katie was cry, we could from simply looking. | 0.9976 |
| | 2 | Josh made me feel uneasiness for the first time ever in my life. | 0.6853 |
| | 3 | Josh feels pity as he paces along to the shop. | 0.6563 |
| | 4 | Terrence feels despair as he paces along to the hairdresser. | 0.6066 |
| | 5 | Ellen made me feel unease for the first time ever in my life. | 0.592 |
| | 6 | Latisha made me feel dismay for the first time ever in my life. | 0.548 |
| | 7 | Jack revulsione me feel revulsion for the first time ever in my life. | 0.5074 |
| | 8 | Frank made me feel revulsion for the first time ever in my life. | 0.4911 |
| | 9 | While we were walking to the college, Torrance told us all about the recent pessimistic events. | 0.4886 |
| | 10 | Roger made me feel unease for the first time ever in my life. | 0.4877 |
| | | **Sentences with The Minimal ATEs** | |
| | **Index** | **Sentence** | **ATE** |
| Factual | 1 | We went to the bookstore, and Alonzo made me feel fearful, really, there is no information here. | 0 |
| | 2 | nothing here is relevant, I made Jack feel angry, time and time again. | 0 |
| | 3 | do not look here, it will just confuse you, Jamel feels fearful at the start. | 0 |
| | 4 | We went to the bookstore, and Justin made me feel irritated. | 0 |
| | 5 | As he approaches the restaurant, Justin feels irritated. | 0 |
| | 6 | Now that it is all over, Andrew feels irritated. | 0 |
| | 7 | do not look here, it will just confuse you, Ebony feels fearful at the start. | 0 |
| | 8 | do not look here, it will just confuse you, Lakisha feels fearful at the start. | 0 |
| | 9 | There is still a long way to go, but the situation makes Lakisha feel irritated, this is only here to confuse the classifier. | 0 |
| | 10 | I have no idea how or why, but i made Alan feel irritated. | 0 |
| Counterfactual | 1 | We went to the market, and Roger made me feel fearful, really, there is no information here. | 0 |
| | 2 | nothing here is relevant, I made Jamel feel angry, time and time again. | 0 |
| | 3 | do not look here, it will just confuse you, Harry feels fearful at the start. | 0 |
| | 4 | We went to the church, and Lamar made me feel irritated. | 0 |
| | 5 | As he approaches the shop, Malik feels irritated. | 0 |
| | 6 | Now that it is all over, Torrance feels irritated. | 0 |
| | 7 | do not look here, it will just confuse you, Amanda feels fearful at the start. | 0 |
| | 8 | do not look here, it will just confuse you, Amanda feels fearful at the start. | 0 |
| | 9 | There is still a long way to go, but the situation makes Katie feel irritated, this is only here to confuse the classifier. | 0 |
| | 10 | I have no idea how or why, but i made Darnell feel irritated. | 0 |

Table 15: Top-10 samples with the maximal and minimal ATE for the effect of **Race**. Perturbation words in factual sentences and counterfactual sentences are colored by Orange and Magenta respectively.

