# OpenReview forum: "Can Transformers be Strong Treatment Effect Estimators? "
_ICLR.cc/2022/Workshop/OSC — Submitted to ICLR2022 OSC _

### Official Review · Reviewer_vaPK · 2022-03-05
**Interesting idea but lacking paper structure**

**Rating:** 1
**Confidence:** 3

**Review:**

This paper proposes a Transformer-based architecture for treatment effect estimation. A common issue for those models is that the impact of the treatment is lost and models can overfit on the covariate. The solution proposed in this paper is to use a Transformer that processes treatment and covariate independently at first, and then allows interactions among them in multi-head attention layers.

While the idea is interesting, the paper is unfortunately written not well. Several parts are missing in the main paper. For instance, Section 2 'Related Work' is left empty. The main text refers two times to this empty section (end of page 1, end of page 3), and it is important for the story of the paper. Further, the model section is incomplete. The Transformer-based architecture is described, but the second listed contribution, an 'adversarial training algorithm for propensity score modeling' is missing. Finally, no experiments are shown in the main paper, although it is mentioned twice that 'comprehensive experiments are conducted'. While some parts can be found in the quite long appendix, it is not acknowledged in the paper, and those parts are essential for the main paper. The paper overall seems like it was written for 10-12 pages, and then sections were removed instead of the aspects of the paper being compressed in 4-5 pages.

Overall, in the current format, the full contributions of the paper cannot be properly evaluated, and the structure of the paper puts it below the acceptance threshold.

---

### Official Review · Reviewer_4B9F · 2022-03-16
**There are no results in the main body of the paper and the related works section is empty.**

**Rating:** 1
**Confidence:** 2

**Review:**

This paper is on topic.
The related work section is missing from the paper, it is just an empty section.
There are no results in the main body of the paper. It is important for a workshop to be able to highlight key results.

---

### Decision · Program_Chairs · 2022-03-24

**Decision:**

Reject

**Comment:**

I agree with the reviewers that the paper in its current form has significant issues that make it impossible to properly evaluate. This is unfortunate because it seems that the paper is quite relevant and has lots of interesting results in the appendix. However, fixing the issues with the main paper requires substantial restructuring and is clearly outside of the scope of a minor revision, so I decided to reject.